# Vision-controlled jetting for composite systems and robots

Thomas J. K. Buchner[1], Simon Rogler[1], Stefan Weirich[1], Yannick Armati[1], Barnabas Gavin Cangan[1], Javier Ramos[2], Scott T. Twiddy[2], Davide M. Marini[2], Aaron Weber[2], Desai Chen[2], Greg Ellson[2], Joshua Jacob[2], Walter Zengerle[2], Dmitriy Katalichenko[2], Chetan Keny[2], Wojciech Matusik[2,3 ✉] & Robert K. Katzschmann[1 ✉]

Recreating complex structures and functions of natural organisms in a synthetic form is a long-standing goal for humanity[1]. The aim is to create actuated systems with high spatial resolutions and complex material arrangements that range from elastic to rigid. Traditional manufacturing processes struggle to fabricate such complex systems[2]. It remains an open challenge to fabricate functional systems automatically and quickly with a wide range of elastic properties, resolutions, and integrated actuation and sensing channels[2,3]. We propose an inkjet deposition process called vision-controlled jetting that can create complex systems and robots. Hereby, a scanning system captures the three-dimensional print geometry and enables a digital feedback loop, which eliminates the need for mechanical planarizers. This contactless process allows us to use continuously curing chemistries and, therefore, print a broader range of material families and elastic moduli. The advances in material properties are characterized by standardized tests comparing our printed materials to the state-of-the-art. We directly fabricated a wide range of complex high-resolution composite systems and robots: tendon-driven hands, pneumatically actuated walking manipulators, pumps that mimic a heart and metamaterial structures. Our approach provides an automated, scalable, high-throughput process to manufacture high-resolution, functional multimaterial systems.

Creating systems that can operate like natural organisms has been a long-standing research challenge[1]. Natural organisms are extremely well adapted to operate effectively and efficiently in their environment. Researchers often tackle this challenge by designing and controlling systems that combine rigid structures with discrete links[4]. These systems are precise (approximately 5 μm)[5], can bear loads (for example, hold up to approximately 14 kg)[6] and can move over uneven terrain[4]. To deploy such rigid systems in the real world, one must use compliant elements at key design locations (for example, at the contact points)[7]. Natural organisms inspire us to widen the design space and introduce soft materials throughout a system's whole structure[2,8]. Hybrid systems that are made of soft compliant materials[9] but contain rigid load-bearing parts[10] can resemble natural organisms at the macroscopic scale[11]. Recently developed hybrid soft–rigid systems can already outperform rigid systems in certain unstructured environments[12,13] by adapting to unknown situations[14] and interacting with living beings in a safe manner[15]. In addition, we must include channels and cavities to carry, for example, signals, power or materials. These features are important but difficult to replicate.

Traditionally engineered systems are precision machined from metal or rigid composite materials. They are hand-assembled and tediously calibrated[4,16]. These systems are usually stiff (greater than 1 GPa)[17] for ease of control; therefore, they contain only a few soft elements or flexible joints. However, hybrid soft-rigid systems are made of polymers that range in stiffness from soft (approximately 3 kPa)[18] to rigid (approximately 3 GPa)[3,19]. They are cast or printed at a coarse resolution with a limited choice of materials[2,20]. An accurate (few tens of micrometers) and rapid (millions of voxels s[−1]) multimaterial additive manufacturing method is required to repeatably produce hybrid soft-rigid systems with a fine resolution at scale.

Direct ink write (DIW) methods produce structures made of multiple resins (that is, epoxy, silicone and nanoclay)[21] with a range of elastic moduli (0.02–1,600 MPa)[22,23] and a resolution greater than 50 μm (ref. 24). Multimaterial multinozzle three-dimensional (3D) printing directly writes from up to eight nozzles (diameter of approximately 205 μm) single lines of materials (approximately 0.225–3,920 MPa) at 40 mm s[−1] (ref. 25). DIW methods support a range of resin viscosities, but scale only proportionally to the number of nozzles and cannot rapidly change between materials on a voxel level.

Traditional 3D inkjet printing uses thousands of individually addressable nozzles to deposit low-viscosity resins that are mechanically planarized and ultraviolet (UV) cured[26]. For a comparable resolution, inkjet deposition leads to orders-of-magnitude faster layer-by-layer printing than other line-by-line printing methods (for example, DIW or fused filament fabrication). Traditional 3D inkjet prints multimaterial bellows that can be assembled to suction grippers[27], intersperses inks to create discrete changes in material stiffness[28], turns soft and rigid acrylates into thin layers of shape memory polymers[29] and jets also non-curing inks to create hydraulic systems[30,31].

[1]Soft Robotics Lab, D-MAVT, ETH Zurich, Zurich, Switzerland. [2]Inkbit, Medford, MA, USA. [3]CSAIL, MIT, Cambridge, MA, USA. ✉e-mail: wojciech@inkbit3d.com; rkk@ethz.ch

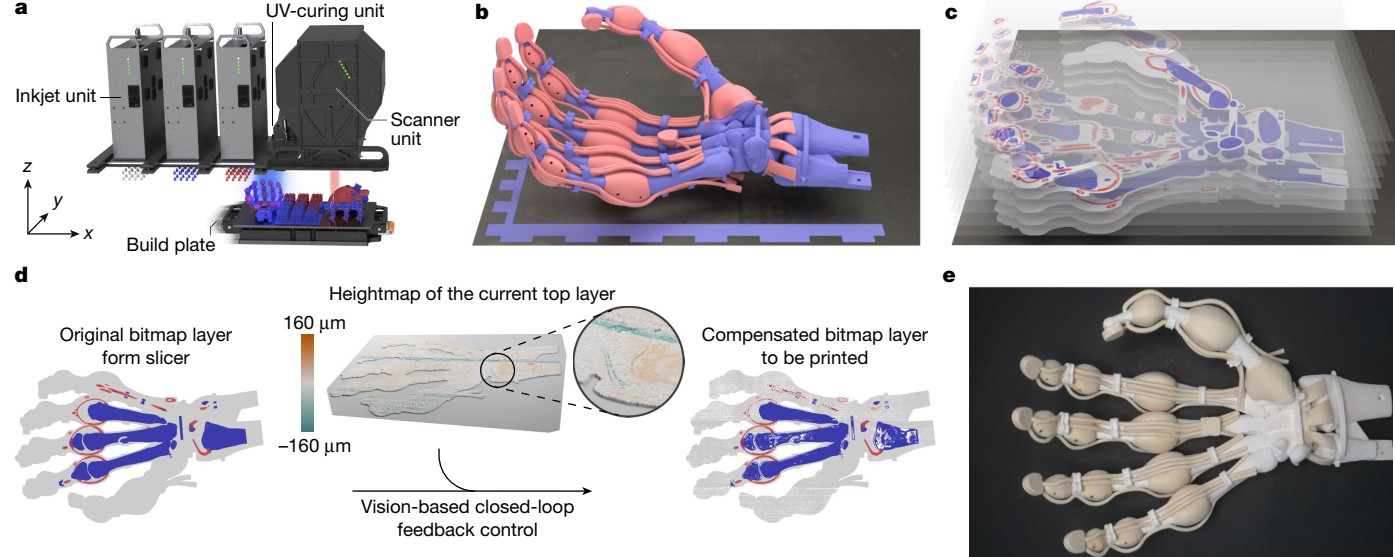

**Fig. 1 | Multimaterial 3D printing of soft and hard materials at a high resolution via vision-controlled jetting. a**, Overview of the 3D printer. A build plate moves underneath the stationary inkjet, cure and scanner units. The multimaterial objects are printed layer-by-layer. For each layer, the inkjet units deposit droplets of ink. The UV curing unit induces the polymerization of the jetted material. The scanner unit records a height map of the deposited layers on the build plate. **b**, A digital 3D model of a complex object to be printed on the build plate. The different materials intended for the object are colour-coded. **c**, An illustration of the stack of layers. The printer's slicer software creates

these bitmap layers from the object's 3D model and required support structures (visualized in grey). **d**, Visualization of the vision-based closed-loop feedback control. The bitmap layers are sent to the printer. Before each layer is deposited, the build plate's surface and material on it is scanned by the scanner unit creating a height map. The control system then modifies the bitmap layer to compensate the uneven surface of the previous layer. The compensated layer is then deposited. This compensation ensures operation in a contactless manner. **e**, A photograph of the printed object after removal of the support material.

Inkjet droplet deposition varies in ink volume due to a variable flow rate and nozzle cross-talk. Therefore, each printed layer requires mechanical planarization, which limits the levels of softness and the type of material chemistries that can be used[32]. Soft or slow-curing materials would be easily smeared and squished by a roller or scraper, leading to material mixing and spatial variation of material composition. Materials only qualify for mechanical planarization if they can be prevented from curing on the roller or scraper. For this reason, 3D inkjet deposition currently only uses acrylate chemistries that rely on fast chain-growth polymerization that only occurs during UV irradiation. Unfortunately, cured soft acrylate chemistries are highly viscoelastic owing to poor control over the degree and structure of their crosslinking. The low printing resolution and the limitation to highly viscoelastic soft acrylates with high hysteresis hinder complex multimaterial robotic designs requiring fine features (Extended Data Fig. 1a–d) and rapidly deforming sections (Extended Data Fig. 1a–c).

A 3D inkjet deposition method called MultiFab[33] relies on slow full-field optical coherence tomography scanning of a small area (2 cm × 2 cm). Each scan has to be rasterized and repeated for the full print surface. All scanned areas are stitched together before corrections to the next print layer can be computed. This leads to a limited throughput of only 0.05 ml min⁻¹ and print times per layer on the order of minutes, even for smaller prints. MultiFab cannot keep up with traditional 3D inkjet owing to the slow deposition speeds and the limitation to acrylate resins that do not scale to produce large functional parts. Especially soft acrylates are not stable to the environment and their deformation behaviour is dominated by hysteresis.

Without access to soft polymers with low hysteresis, it is not possible to reproduce complex functional materials and structures with desirable properties. Printing hybrid soft-rigid systems necessitates functional polymers that can crosslink in a controlled manner to minimize viscoelasticity while achieving a wide range of stiffnesses. Complex functional systems also require cavities and channels of fine resolution across large build volumes despite a high print throughput. These

desirable material chemistries and structural features can be realized if we employ a non-contact planarization strategy and allow for easily removable support materials (such as wax) (Extended Data Fig. 2).

## Vision-controlled jetting

Here we present a method for inkjet-based multimaterial deposition using contactless, continuous print adjustments. The method expands the range of printable materials and the degree of material hardness to create functional complex systems and robots (Supplementary Video 1). We call this manufacturing method vision-controlled jetting (VCJ) (Fig. 1 and Supplementary Video 2). Our method utilizes a high-speed 3D vision system to capture a depth map of the currently printed surface, and it compensates for deviations from an ideal planar surface by locally adapting the amount of resin to be jetted in the next layer. The method's vision system uses four cameras and two laser sources for laser profilometry while printing. The feedback loop including the surface scan of the whole print area is performed without slowing down the print process. Our method is 660 times faster than previous work[33] by achieving a throughput of up to 33 ml min⁻¹.

Our contactless procedure enables us to print chemistries that cure slowly, such as thiol-enes[34–36], which polymerize via a step-growth process. This slow cure mechanism builds the polymer's structure with more precision, which provides us with control over both the polymer's backbone and the degree and structure of crosslinking (Fig. 2). Acrylate resins, used in conventional 3D jetting, have only a random distribution of crosslinking polymer backbones and are therefore much less controllable in their curing status. Using VCJ on slow-curing chemistries enables the fabrication of parts with a wide range of material properties (Extended Data Table 1), including chemical and outdoor (UV and moisture) compatibility (Extended Data Fig. 3).

Our printer can create hybrid structures at a high resolution (32 μm × 64 μm × 20 μm voxel size) and with a high throughput (24 × 10⁹ voxels h⁻¹) for build speeds of up to 16 mm h⁻¹ in the *z* direction for a

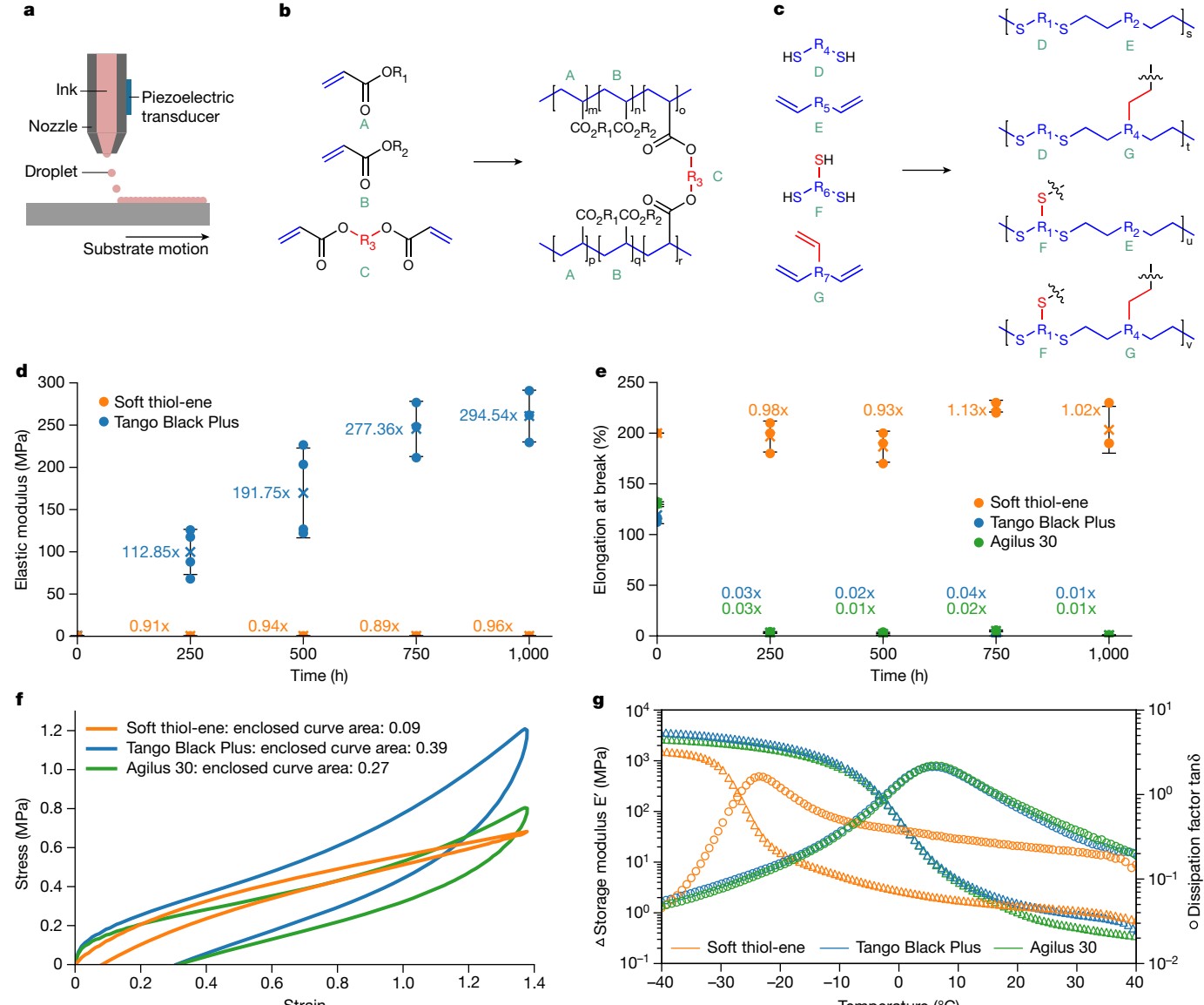

**Fig. 2 | Material characteristics compared to the state-of-the-art. a**, Schematic of 3D material jetting. Jetting describes a process where ink in a nozzle is ejected in the form of droplets onto a substrate using a piezoelectric transducer **b**, Acrylate chemistry uses functional monomers cured via radical polymerization (represented by molecules A, B and C) in their inks. Chain-growth polymerization results in a random and irregular distribution of monomers along the polymer backbone (blue). Given that all acrylate polymers, rigid and soft, contain this common backbone chain (blue), the properties are controlled by the monomer side groups, $R_1$ and $R_2$, and both the structure and degree of crosslinks $R_3$. This combination of structural features typically results in a viscoelastic polymer. **c**, Vision-controlled jetting has enabled printing of a step-growth polymer (for example, thiol-ene) using radical polymerization. The polymer chain consists of alternating thiol (D or F) and ene (E or G) monomers. The properties of the polymer are defined by the regular backbone structure and the degree of crosslinking. **d**, The change in elastic modulus during outdoor weathering for Tango Black Plus and soft thiol-ene. Error bars indicate one standard deviation from the mean over four samples. **e**, Change in breakdown strength during outdoor weathering for Tango Black Plus, Agilus 30 and soft thiol-ene. Error bars indicate one standard deviation from the mean over three samples. **f**, The hysteresis of the material is defined by the area enclosed by a stress-strain cycle. The two acrylates tested showed 3- or 4.43-fold increased hysteresis. **g**, A dynamic mechanical analysis (DMA) was conducted on soft thiol-ene and two acrylates (Tango Black Plus and Agilus 30). Changes in storage modulus were significant between −35 °C to −18 °C, whereas the two acrylate samples showed significant changes in storage modulus in a much wider range between −30 °C to 15 °C.

total of $6.15 \times 10^{11}$ individually assigned discrete 3D volume elements (voxels). The printer's build volume is 500 mm × 245 mm × 200 mm. The print heads and the print speed are therefore on-par with state-of-the-art inkjet printers for this scale and resolution but our print system including the scanner allows for the first time a wider range of chemistries and therefore material properties compared to other printers.

We can use both fast and slow chain-growth polymer chemistries as resins with our contactless approach. Droplets of different types of resins (build material) are jetted concurrently with droplets of wax (support material). The resin then polymerizes using UV-radiation and the wax solidifies when cooled down. Like conventional inkjet printers, our process also utilizes piezoelectric print heads (4 print heads per material; 1,024 nozzles per print head), but we additionally introduce a fast surface profilometer. The closed-loop controlled system (Fig. 1d) allows for single-pass multimaterial printing with currently up to three build materials and one support material (Extended Data Fig. 1g–i show three materials).

The 3D vision system provides highly accurate depth maps of the surface at a resolution of 64 μm × 32 μm × 8 μm. The depth maps are captured and processed without slowing down the print. A laser triangulation method focuses a line laser on the surface, and a set of high-frame-rate cameras captures the images as the print bed moves under the scanner. This scan generates data at 2 GB s$^{-1}$. The system uses a high-performance graphics processing unit to convert the image data to a 54-megapixel depth map. This computation takes less than one second so the control feedback loop can immediately compare the depth map to the sliced computer-aided design model of the parts and adjust the printing volume density per voxel. The areas with excess material receive fewer droplets, and the areas with insufficient material receive a higher density (up to the maximum deposition rate).

The contactless printing process allows us to resolve fine multimaterial features (Extended Data Table 2) and to print not only slowly curing step-growth polymers (such as thiol-enes, Extended Data Fig. 1a–c and Extended Data Table 1a,b), but also continuously curing polymers (such as UV-cationic cured epoxies, Extended Data Fig. 1d–f and Extended Data Table 1c,d). Different materials within the same family can be arbitrarily arranged on a volumetric grid. Using our thiol-ene material family, we can print highly elastic objects that maintain mechanical properties over time (Extended Data Fig. 3). The process also allows us to use phase-change waxes as a support material, without the risk of smearing and mixing the build and support materials. Our wax melts into a drainable fluid with a low viscosity in the postprocessing step (Extended Data Fig. 2 and Supplementary Video 3) to deliver key geometries such as long channels (up to 45 cm) with small diameters (greater than or equal to 750 μm) (Extended Data Table 2). Moreover, the high resolution of the material deposition system allows for wax-based separators with the thickness of a single voxel between different material chemistries (Extended Data Fig. 1g–i).

## Material characterization

We evaluated the properties of the materials and compared them to state-of-the-art 3D jetting materials, namely, acrylate resins (Fig. 2). Specifically, we compared soft thiol-ene to Stratasys PolyJet material Tango Black Plus. When new, their elastic modulus and elongation at break match closely. We tested the change of elastic modulus of acrylate (Tango Black Plus) and soft thiol-ene when exposed to outdoor weathering including UV exposure, temperature changes and humidity (ASTM G154). The elastic modulus of Tango Black Plus (0.89 MPa) had increased after just 250 h by approximately 113-fold and after 1,000 h by approximately 295-fold to 261 MPa. In comparison, after 1,000 h only an approximately 4% change in elastic modulus (0.53 MPa) was observed for our soft thiol-ene (Fig. 2d and Extended Data Fig. 3). Our soft thiol-ene's elongation at break changed by less than 1.13-fold compared to when new. In comparison, the acrylate-based Tango Black Plus turned brittle by 0.03-fold within less than 250 h of weathering at a reduced elongation at break of approximately 3% from initial 119%. The acrylate Agilus 30 by Stratasys, the successor of Tango Black Plus, also showed a similar behaviour (Fig. 2e). The modulus of resilience (ASTM D2632) directly after printing was 7% for Tango Black Plus and 14% for Agilus 30, while soft thiol-ene was double at 27%.

The viscoelastic behaviour of the materials was quantified using stress-strain cycles with up to 140% displacement. The acrylates' hysteresis area is 3 to 4.3 times larger than that of thiol-ene's (Fig. 2f). Tango Black Plus had an area of (0.42 ± 0.05) MPa, Agilus 30 an area of (0.29 ± 0.03) MPa, while soft thiol-ene only had an area of (0.087 ± 0.006) MPa. A dynamic mechanical analysis (DMA) of soft thiol-ene (Extended Data Fig. 4) and of the acrylates showed that soft thiol-ene had a much narrower region of glass transition $T_g$ compared to the two acrylates (Fig. 2g). Soft thiol-ene's changes in the storage modulus were significant between −35 °C to −18 °C, whereas the two

acrylate samples showed significant changes in the storage modulus within a much wider range, between −30 °C to 15 °C (Fig. 2g).

## Printed systems and robots

We show the new capabilities of our printing process through several functional multimaterial systems that were inspired by nature. After printing, our system only requires dissolving the support structure (Supplementary Video 3), connecting pneumatic supply lines and, in some cases, sealing the support's drainage holes. In summary, we present a robotic hand, a walking robot, a robotic heart and a metamaterial structure. The tendon-driven hand derives from the magnetic resonance imaging data of a human hand[37], and it has contact sensor pads at the fingertips and on the palm. The walking robot locomotes with six legs, senses contact with the environment, and manipulates objects with its manipulator arm. The shape of the fluidic pump and its integrated valves were inspired by a mammalian heart. Finally, the truss-like metamaterial allows for preprogrammed changes in the material stiffness. We describe the resulting systems in the following.

**Tendon-driven hand.** Humans smoothly interact with objects and the environment predominantly through touch by hand[38]. Therefore, researchers have been using 3D printing to develop touch sensors[39], sensorized skin[38,40] and sensorized grippers[41]. Our multimaterial tendon-driven hand is a fully functional print that is fitted with sensor pads and pneumatic signal lines (Fig. 3a,b and Supplementary Video 4). The hand can sense contact and initiate grasping and stop finger motion when the fingertip contacts a grasped object. These skills are made possible by VCJ's ability to print long, soft, and thin channels as well as large cavities with thin membranes. The 19 independently actuatable tendons in our bioinspired hand were designed with a rigid load-bearing core and a soft bendable shell (Fig. 3b). Our printing method allowed us to manufacture this complex system much more easily than other anthropomorphic hands[42].

We connected a set of tendons to servomotors for actuation. The fingertips are equipped with print-in-place sensor pads, which are connected to pressure sensors through printed fluidic signal lines (Fig. 3b). When the hand contacts an object and the sensed pressure exceeds a threshold value, control actions on the servomotors are triggered (Extended Data Fig. 5). For example, when the palm sensor touches an object, it triggers a grasping action (that is, all the fingers bend). When an object contacts a finger's sensor pad, the change in pressure is detected through the signal lines. Once a predetermined threshold in pressure is reached, the finger motion is stopped before it reaches a full curl (Fig. 3c). We also actuated individual fingers of the hand, for example the opposable thumb can touch the tips of the other fingers (Fig. 3d). This sensorized setup allows the hand to autonomously grasp objects. The hand's grasping capability was evaluated using a set of objects (Fig. 3e).

**Locomoting gripper.** There are robots that locomote and react when they contact the environment[41,43,44]. However, these impressive examples require several manufacturing methods and complex manual assembly. After support removal, our printed fluid-driven walking robot (Fig. 4a and Supplementary Video 5) can locomote (Fig. 4f–h), grasp (Fig. 4i) and sense (Fig. 4i). These capabilities were made possible by the printer's ability to create strong airtight soft-rigid interfaces and complex 3D channels (Extended Data Fig. 6a). Also, the elasticity and low damping of the material allow the robot to move quickly and render it easy to control.

Our directly printed walker has a six-legged arrangement that is fitted with dual-joint legs (Fig. 4b and Extended Data Fig. 6a). Each leg has a range of motion from 0° to 30° at the upper joint and from 0° to 20° at the lower joint; each joint supports actuation pressures up to 35 kPa. The six legs are supplied with pressurized air in two groups of three legs (Fig. 4c). There is one supply line for the upper joint actuator and one for the lower joint actuator (Fig. 4b,c). In addition, the robot is fitted

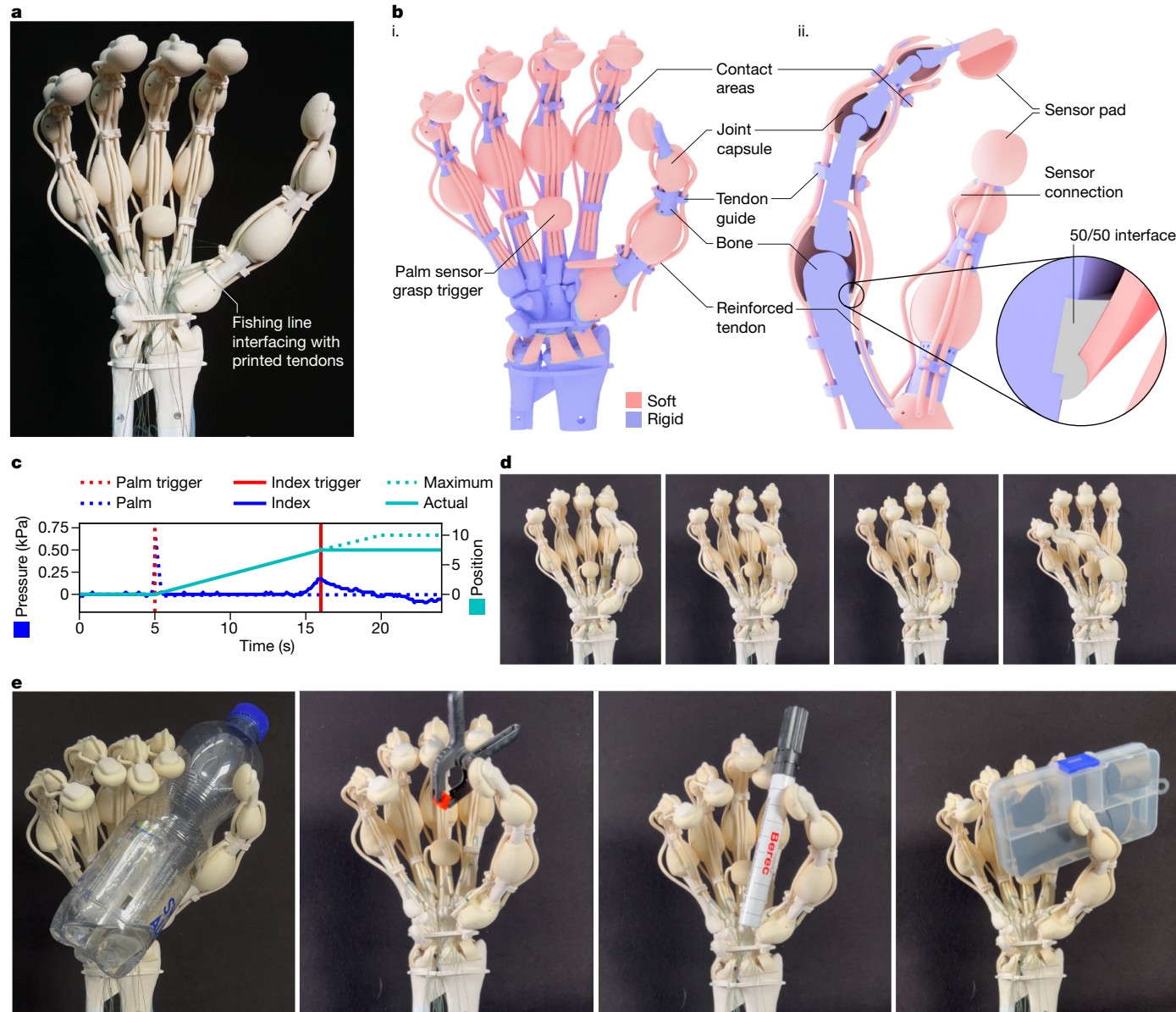

**Fig. 3 | A printed robotic hand, sensorized and driven through tendons.** **a**, A tendon-driven five-fingered soft-rigid hybrid gripper imitating a human hand printed in one process. **b**, (**i**) Rendered representation of the gripper with rigid thiol-ene parts shown in blue and soft thiol-ene parts in pink. (**ii**) Cross-section of the index finger with tendon structure and sensor cavity as well as joint geometry visible. The 50/50 interface part comprises a mixture of the two materials. **c**, Visualization of the grasping sequence. After 5 s, the palm pressure sensor triggers the start of the grasping motion. The index finger starts to close, visualized by the change in position. At 16 s, the index finger triggers the stopping of the grasping motion before the maximum position would be reached. **d**, The reachability of the hand is visualized in a series of images showing the ability of the hand to make contact between the thumb and each finger. **e**, Different objects grasped employing a control algorithm that starts the grasp at contact with the palm sensor pad and closes the hand around an object until the maximum position is reached or contact with an object is detected.

with a gripper that has embedded sensor pads at its tips (Fig. 4b,c). The gripper's arm can move up and down using its upper and lower actuators, and the gripper can grasp and lift objects (Fig. 4i). The sensor pad in the tip of the gripper (Fig. 4b,c) provides feedback on grasping when contact is established (Fig. 4e,i); the sensor feedback informs the controller how to adapt the robot's action. The walker is also able to change the direction it is facing by turning (Fig. 4f) at a speed of $(20/15)° s^{-1}$. Our chosen gait cycle (Fig. 4d,h) allows the robot to locomote at a speed of approximately 0.1 (body length) $s^{-1}$ or approximately 0.01 m $s^{-1}$ in a stable manner (Fig. 4g and Extended Data Fig. 6b). In addition, reversing the gait cycle moves the robot backwards.

**Fluid pump inspired by a heart.** We also printed (Fig. 5a) a fluidically driven pump designed to resemble a heart (Fig. 5b and Supplementary Video 6). This pump has actuation membranes, one-way valves and internal sensor cavities embedded in the heart's chamber (Fig. 5b). The integrated valves and pumping membranes were inspired by the geometries and mechanisms in mammalian hearts, which have already been optimized by nature. Our easy-to-remove support material (Extended Data Fig. 2 and Supplementary Video 3) allowed us to print several small and large cavities with thin, soft membranes and rigid walls in one process (Fig. 5b). Similar pump designs were previously only possible through the casting or injection moulding of individual components, both of which were followed by time-consuming and labour-intensive assembly[45–47].

The pumping cycle of the bioinspired pump is controlled by the inflow and outflow of air into the actuation chamber. The cyclic

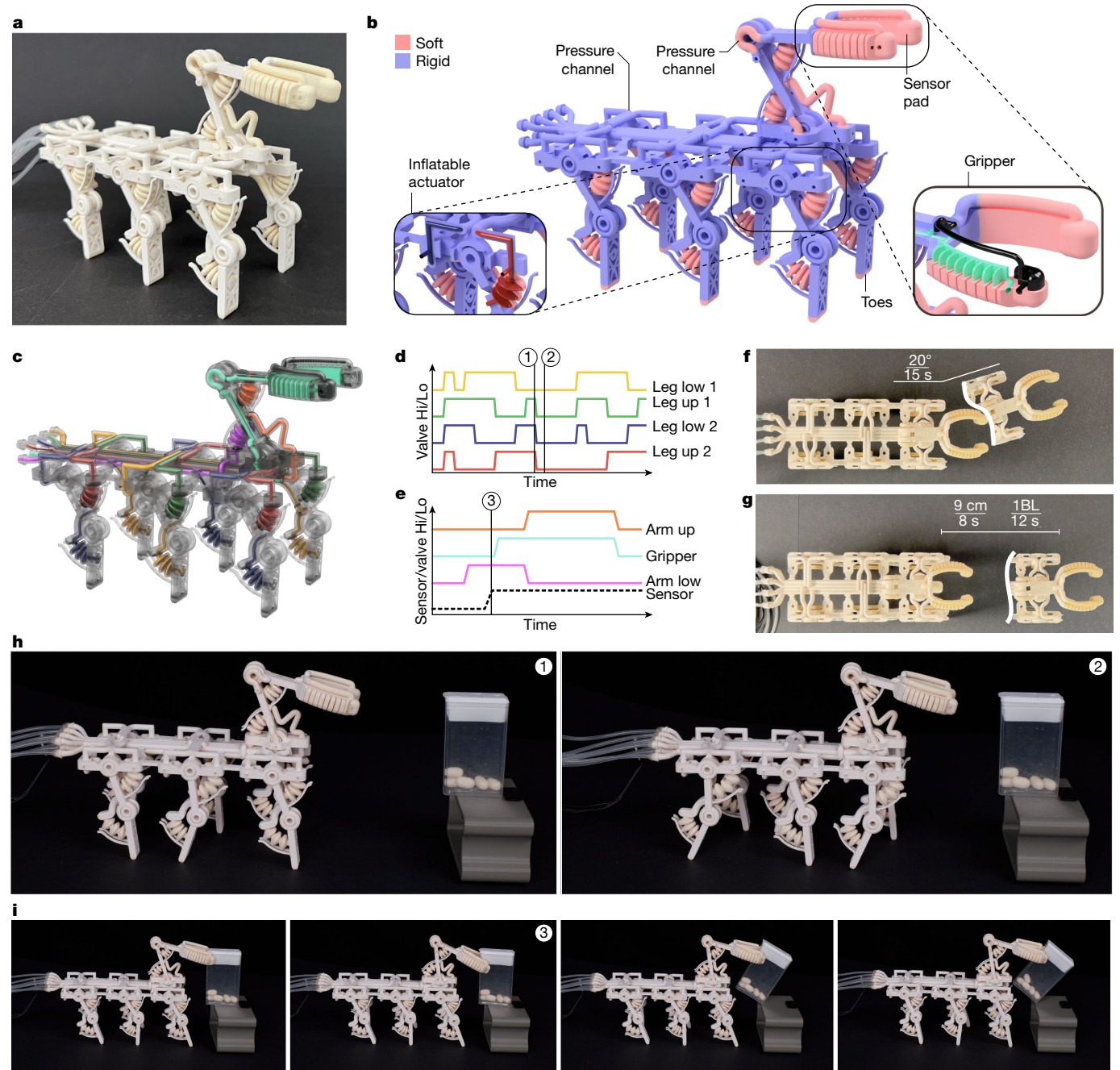

**Fig. 4 | A locomoting, sensing and grasping robot that is functional after 3D printing. a**, A pneumatically actuated soft-rigid hybrid robot printed with an all-in-one process. **b**, Rendered illustration of the robot with rigid parts shown in blue and soft parts visualized in pink. A zoom-in onto the gripper element is shown with the chamber structure visible (green). The sensor pad's cavity and signal line are shown in black. The two-jointed leg of the robot is in the relaxed position with the upper actuator's channel structure in red and the lower channel in blue. **c**, An illustration of the four pneumatic networks used for the locomotion of the robot. The colour code for the channels is shown in **d** for the channels related to gait and in **e** for grasping. **d**, Actuation pattern of the four chambers used for locomotion. During initialization, all actuation channels are pressurized. After initialization, the gait cycle starts once 'Leg low 1' is pressurized for a second time. Two states of the gait indicated by the vertical lines labelled '1' and '2' are illustrated in **h**. The gait cycle repeats once 'Leg low 1' is pressurized again for a third time. **e**, The actuation patterns for a grasp are indicated with solid lines and the pressure in the sensor pad is indicated by a dashed line. The closed gripper state is indicated by the vertical line '3'. The control logic would depressurize the gripper system if the pad's sensor signal is below a threshold (that is, contact is not detected). Given that contact is detected for the example in '3', the gripper is commanded to lift the grasped object. **f**, Visualization of the robot's ability to turn. **g**, The robot's speed when locomoting. **h**, Still images of the gait states '1' and '2' as illustrated in **d** with the change in leg position visible. **i**, Sequential still images of the grasp procedure with contact detection in '3', as illustrated in **e**.

change of the actuation chamber pressure repeatedly deforms the actuation membrane, which in turn leads to the intended flow of liquid (Fig. 5c). The mechanism design of the multimaterial valves was inspired by nature and further optimized in its arrangement of soft and rigid materials and feature dimensions. The different steps in the multimaterial valve optimization process (Fig. 5d) were made possible due to the fast prototyping ability of the multimaterial 3D printer.

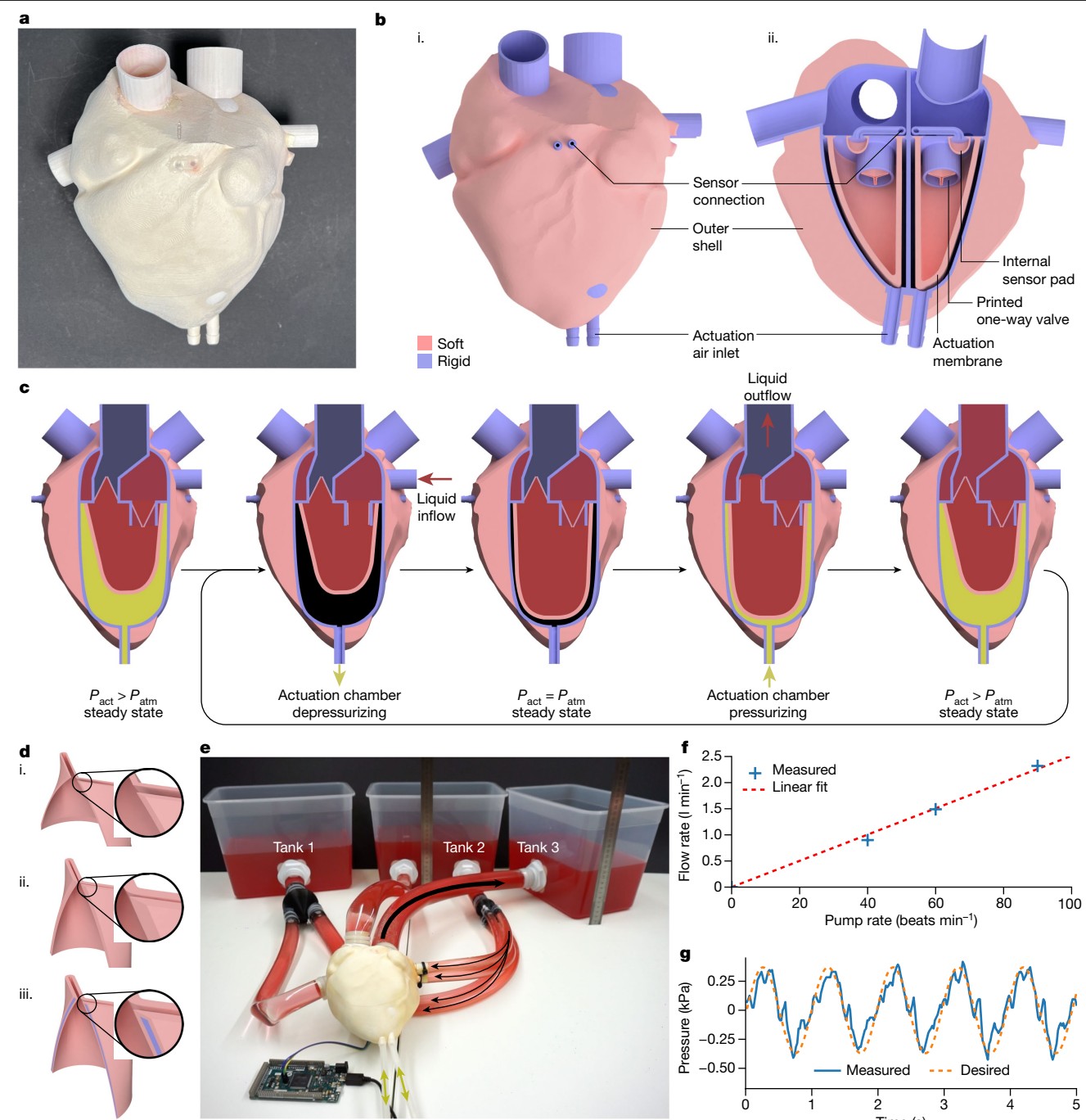

**Fig. 5 | Functional heart pump, printed in a single process. a**, A mammalian-heart-inspired soft-rigid hybrid pump printed in one process. Two of its four chambers are fluidically actuated and equipped with sensors. **b**, Illustration of the pump (**i**) with rigid thiol-ene parts shown in blue and soft thiol-ene parts visualized in pink. (**ii**) Cross section cut of the heart showing the internal sensor cavity, the position of the one-way valves, the actuation membrane and the inlet for air actuation. **c**, Sequential still images for the pump cycle showing a sketch of the pump's cross section inside view. The cyclic change in actuation pressure $P_{act}$ compared with atmospheric pressure $P_{atm}$ and the resultant liquid flow are indicated. **d**, Three design iterations of the printed valve. (**i**) A thin, soft membrane that was prone to invert on itself under pressure. (**ii**) A thicker membrane that was still not strong enough to avoid inversion. (**iii**) A functional membrane with internal, rigid reinforcement. **e**, Heart pump connected to three liquid tanks. This setup was inspired by the mammalian cardiovascular system. A microcontroller is used to read the sensor data and a piston setup is used to provide actuation pressure. **f**, The pump setup can move liquid in relation to the applied pump rate. A maximum flow rate of 2.3 l min$^{-1}$ at 90 beats min$^{-1}$ was measured. **g**, The correlation of the commanded signal to the sensed chamber pressure at 60 beats min$^{-1}$.

We tested the pump's flow rate, sensors and ability to retain water with a fluidic setup (Fig. 5e). The flow rate was measured for different pump cycles ranging from 0 beats min$^{-1}$ to 90 beats min$^{-1}$, resulting in flow rates of up to 2.3 l min$^{-1}$ (Fig. 5f). The printed sensors measure pressure changes matching the desired pumping frequency; therefore, they could be used in a closed-loop controlled setting (Fig. 5g).

**Metamaterial structures.** Nature not only shows complex combinations of soft and rigid materials like in the heart but is also abundant in materials with a wide range of properties. For example, materials can

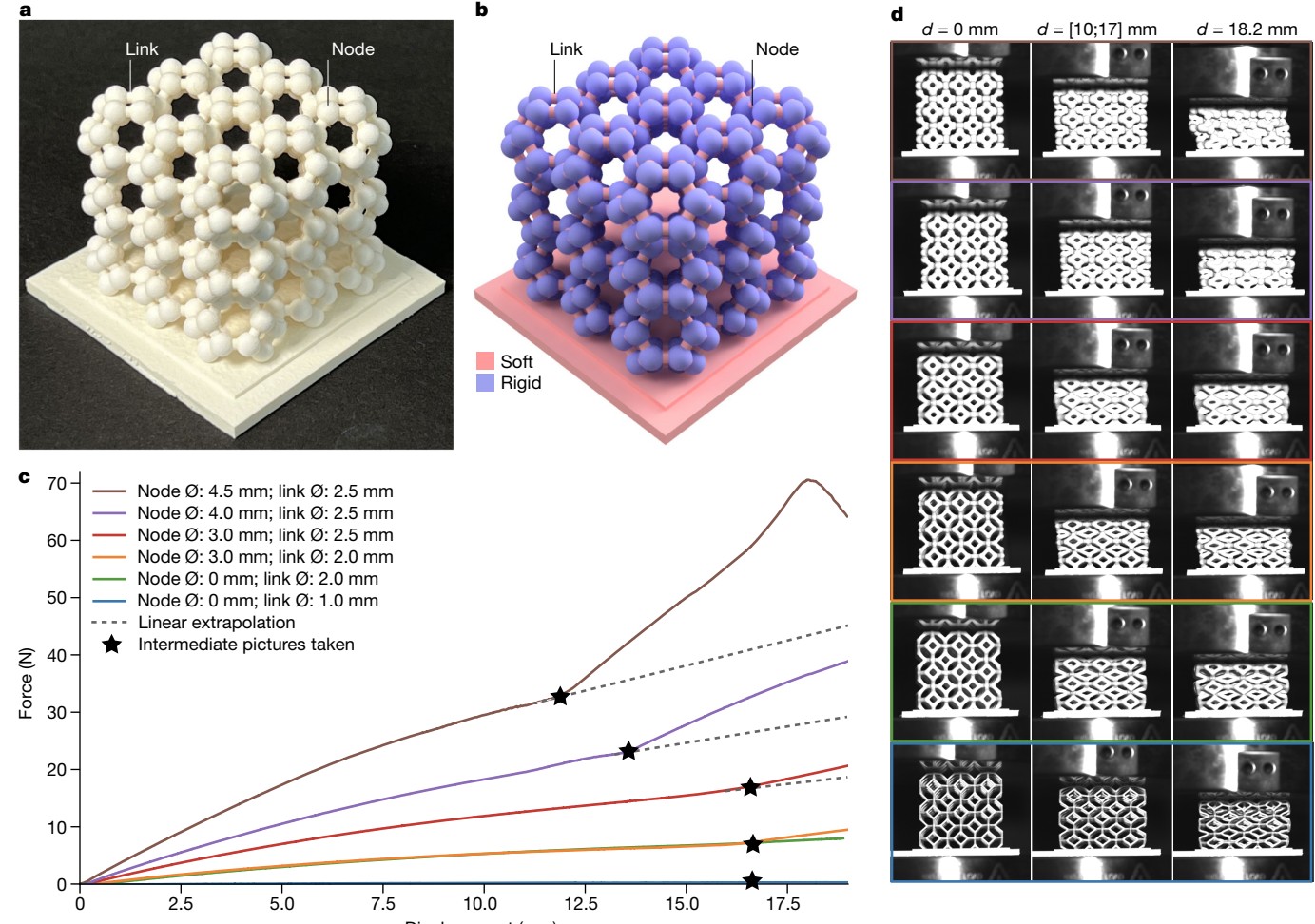

**Fig. 6 | 3D printed multimaterial metamaterial structures with tunable changes in stiffness. a**, 3D structure, 3 × 3 × 3 cells with constant diameter of the links and nodes, respectively. **b**, The links rendered in pink are soft, and the nodes rendered in violet are rigid. **c**, Force displacement curves for compression tests on six different specimens. The decrease in force at high displacement in the brown plot line corresponds to the collapse of the structure due to buckling. The bend in the force strain curve in the second half of the displacement graph corresponds to the rigid nodes making contact, leading to increased stiffness of the structure. The dashed lines show a prediction of how the curve would have behaved without the rigid nodes making contact. **d**, Still images of compression tests conducted on the material. No compression ($d$ = 0 mm) is shown to the left; compression when the nodes are touching ($d$ between 10 mm to 17 mm) is shown in the centre; and approximately 50% compression is shown to the right.

have various trade-offs in stiffness and weight. VCJ can produce material architectures that go beyond what we know from nature. For example, metamaterial structures supersede a material's properties because of their topological design. The design of metamaterials uses various physical principles, allowing a designer to tailor these structures to specific tasks[48]. Structural metamaterials are currently predominantly 3D printed[49,50]. While traditional metamaterials are constructed from single materials, material systems that combine materials with diverse properties could lead to a wider field of application.

To demonstrate a metamaterial architecture using VCJ's material chemistry and fine features, we printed a range of truss-like metamaterials (Fig. 6a). Here, we manufactured a set of 3 × 3 × 3 unit cells that comprised geometrically oriented links (diameter of 1.0 mm to 2.5 mm) and nodes (diameter of 0.0 mm to 4.5 mm). Varying the diameter of the soft links and rigid nodes (Fig. 6b) changed the metamaterial's behaviour under compression. We were able to fabricate these delicate structures at such low effective densities because of the printer's high resolution and its support material, which melts away easily. We investigated our metamaterial's change in behaviour by performing compression tests on different samples (Fig. 6c). We were able to tune the behaviour under compression by changing the diameter of the links and nodes (Fig. 6c). The amount and the onset of the sudden change in the material property was mostly influenced by the diameter of the nodes. For one configuration, the structure buckled for compression greater than or equal to 17.5 mm (Fig. 6c,d, brown plot).

## Discussion

As the examples above illustrate, we have developed an automated, high-throughput approach to manufacturing high-resolution, durable multimaterial functional systems in a single fabrication process. Our results illustrate that this new printing method can create complex multimaterial functional robots with integrated sensing and actuation channels. The printer's high resolution, speed and wide range of material properties enable a new set of hybrid soft-rigid robots. The printer can use a wider range of possible material chemistries, which allows us to build functional and long-lasting materials. Our contactless printing approach can now create geometries with almost any internal structure, such as inner cavities, 3D fluid flow channels, tendon guides and pressure sensing lines. Having freeform control of how the soft and rigid materials are placed at the voxel level within

a design drastically improves the functionality and performance of printed multimaterial systems.

The VCJ printing technology widens the palette of available materials but is still limited by a relatively low viscosity of the UV-induced curing materials. Exposing these materials to outdoor weathering led to small deformations of some samples, for example, warping (Extended Data Fig. 3e–h). This warping led to an increased variation in the tested properties. Another challenge was that the interface of prints from different material chemistries did not always adhere well, yet specific tuning of the chemical composition of the materials can further improve that in the future. As a workaround, the high resolution of the printer allows for the printing of features for mechanical interlocking of multimaterial interfaces. The availability of only four print heads in the current design still limits the complexity of multimaterial designs that can be printed in one process. The print process is inherently based on the ample use of support material. While the support material is easily liquefied and removed (Extended Data Fig. 2 and Supplementary Video 3), every created cavity must have a connection to the outside of the printed part for drainage. In particular, the removal of the wax support material from small cavities or porous sponge-like structures is difficult despite the use of surfactants to lower the high surface tension.

We anticipate that VCJ will open new possibilities to quickly and repeatably create complex objects or machines that were previously impossible to produce. Our freeform fabrication technology widens the design space that is available to engineers and scientists so that we may rapidly create hybrid soft-rigid structures, systems and robots at the millimetre to decimetre scale. Our rapid and versatile manufacturing technology will create new opportunities for scientific investigations, experimental design, complex prototyping and industrial innovation.

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

# Methods

The results of this work were created with our contactless manufacturing system, which allows for a high print throughput independent of the structure that is to be printed. Our method allows us to place voxels of materials in freeform. The support material can be melted and washed away easily to allow for the creation of functional channels, cavities and hollow structures. In the following section, we describe the Vision-Controlled Jetting method and the evaluation methods that we used on our printed structures, systems and robots.

## Vision-controlled jetting

The examples presented in this work were all 3D printed using a multimaterial additive manufacturing platform that utilized a vision-controlled jetting technology (Fig. 1 and Supplementary Videos 2 and 3). The platform has a scanning system, jetting system and positioning system that can now employ suitable material technologies, produce accurate print results and scale up in terms of size and throughput. The platform is composed of six subsystems described in detail in the following:

1. The positioning subsystem moves the build plate to a certain location according to the commands that are issued by the print control software at a set velocity and along a set path.
2. Each of the inkjet units contains four print heads (three units are shown in Fig. 1a), drive electronics, material feeds and a pressure control system to jet a specific material onto the build plate.
3. The UV curing unit uses UV LEDs to cure the materials that have been deposited onto the build plate.
4. The scanner unit uses its laser profilometry system to generate a high-resolution topographical map of the build surface.
5. The print control software steers the printing system's processes by utilizing the scanner data to generate adapted print layers as needed.
6. Postprocessing removes the support material from the completed prints (Extended Data Fig. 2).

**Positioning subsystem.** The positioning subsystem controls the location of the build plate relative to the rest of the printer. An axial motion system utilizes a linear motor to move the build plate under the print hardware along the $x$ axis. The $y$ and $z$ axes are driven by brushless DC motors. The $y$ axis is used to shift the location in which the structure is built in such a way that it is aligned relative to the print head array. This axis allows the control loop to compensate for variations in nozzle performance. The $z$ axis ensures that the build surface stays within working distance of the print heads as the build progresses. All the axes are operated as servos, using encoders with a 1 μm resolution for position and velocity control. The print velocity is limited by the deposition frequency and the resolution.

**Inkjet units.** Each inkjet unit contains all the hardware and electronics that are required to print a single material. Each unit contains four print heads (Fujifilm Dimatix SG1024-L), which are placed in a staggered array to fully cover the build plate. This allows a complete layer to be deposited each time the build plate passes under the inkjet units. The print heads have a native resolution of 400 dots per inch (DPI), and they eject droplets with a volume of about 70 pl at 15 kHz. The drive electronics translate the requested layer data into firing pulses for the nozzle actuators while the build plate is scanned under the print heads.

**UV curing unit.** After the deposition of a layer, the build plate moves under the curing unit (UV LED lamp) to initiate the polymerization of the printed material. One lamp is present on each side of the inkjet units to allow for bidirectional printing. The lamps emit 405 nm light at 16 W cm$^{-2}$.

**Scanner unit.** The scanner unit uses custom laser triangulation profilometry. A laser line is projected onto the surface of the build plate as it is passed under the scanner. An imaging system reads the shape of the laser line from a 32 × 2,048 laser line image, and it computes a two-dimensional height map with 2,048 pixels at each sampling interval. Each camera in the imaging system captures 6,000 laser line images per second. Four cameras are needed to cover the full width of the build plate, and each camera captures 9,000 images per scan to cover the length of the print. The two-dimensional height maps obtained from the individual cameras are assembled into a full 3D height map of the build surface. In total, each height map is computed from $2.36 \times 10^9$ pixels within 2.5 s. The height maps are geometrically calibrated to a fixed pixel resolution of 32 μm × 64 μm × 20 μm and are provided to the feedback control system.

**Print control software.** The print control software orchestrates the activity of each subsystem to execute successful prints. When a build job has been defined, the print control software processes the file in a voxel representation by rendering the geometries at the printer's resolution[51,52]. The voxelization step uses ray tracing to quickly render the input geometries.

To print a layer, a feedback algorithm generates the layer data based on the scan data from the previous layer and from the input geometry's voxel representation[53,54]. The feedback algorithm aims to maintain the printing plane at a fixed distance from the scanner by reacting to the height of each voxel in the scan data. If the print is higher than the desired level, the feedback control can reduce the amount of ink that is deposited or skip printing at that voxel in the next layer. If the print height is too low for a given voxel, the feedback control will determine which material is missing according to the currently measured height. It will then increase the amount of ink that is deposited up to the maximum capacity of the print head.

The generated layer command is sent to the drive electronics of the print head. The drive electronics deposit the materials into their desired positions while the motion control system moves the build plate underneath the print hardware. This process is repeated as the parts are built up layer-by-layer until the build has been completed.

**Postprocessing the prints.** Completed builds are encased in a support material, which must be removed before the parts are ready for use (Extended Data Fig. 2 and Supplementary Video 3). The entire build is first placed in a convection oven and heated to 65 °C, where it is left overnight for the bulk of the support material to melt and drain away. The parts are then removed and placed in a tank of cleaning solution and heated to 65 °C, where they are sonicated for 20 min. The parts are then rinsed with water and allowed to dry in air. The drainage holes in the printed parts for this study are sealed using cyanoacrylate.

**Print velocity.** The inkjet units cover the print bed along the printer's $y$ direction (Fig. 1a). The print bed moves back and forth underneath the inkjet units in the $x$ direction at velocity $v_x$ (equation (1)). The velocity in the $x$-direction is dependent on the jetting frequency $f_{jet}$ of the print head's nozzles and on the resolution in the $x$ direction $r_x$ (equation (1)). The minimum droplet size determines the resolution $r_x$ (here, 32 μm) and the actuation speed of the print head's piezo nozzles limits the jetting frequency $f_{jet}$. The jetting frequency is adapted for each material to ensure best print performance.

$$v_x = r_x f_{jet} \tag{1}$$

Due to the exothermic nature of the curing process, the printed part is cooled for a certain amount of time $t_{cooling}$ after each layer has been deposited. Therefore, the printer takes the time $t_{layer}$ to print a single layer of a certain length (in the $x$ direction) $l_x$ and width (in the $y$ direction) $l_y$. The length $l_x$ is determined by the size of the full print bed and

the distance that spans across all the inkjet heads, the UV lamp and the scanner. Since the print heads cover the whole print bed in the $y$ direction, the layer time $t_{layer}$ does not depend on the $y$ extension of the print bed (equation (2)).

$$t_{layer} = l_x/v_x + t_{cooling} \qquad (2)$$

The height of each layer $h_{layer}$ can be adapted, and it is dependent on the total number of deposited droplets in each $x$ location. The layer's height is also dependent on the volume of the jetted droplet $V_{droplet}$ and the resolution in the $x$ direction $r_x$ (here, 32 µm) and $y$ direction $r_y$ (here, 64 µm). The print head's speed in the $z$ direction $v_z$ determines the overall print speed (equation (3)). In contrast to other printing methods, the speed for this type of inkjet deposition system does not depend on the printed object's geometry in the $y$ direction. The speed is, however, dependent on the resolution in the $x$ direction $r_x$ and the resolution in the $z$ direction $r_z$ (here, 20 µm), that is, it depends on the layer's height $h_{layer}$ (equation (3)).

$$h_{layer} = V_{droplet}/(r_x r_y) =: r_z$$
$$v_z = r_z/t_{layer} \qquad (3)$$

Inserting equation (1) and equation (2) into equation (3) describes the relation of resolutions to print speed (equation (4)).

$$v_z(r_z, r_x) = r_z/(l_x/(r_x f_{jet}) + t_{cooling}) \qquad (4)$$

The user can adjust the print velocity $v_z$ (here, 16 mm h$^{-1}$) by adjusting the jetted droplet's volume. The droplet's volume can be tuned by adjusting the fluid's rheological characteristics or by changing the print head's operating parameters (such as the piezo actuation waveform or jetting temperature). The total print duration is determined by the build job's width in the $x$ direction and height in the $z$ direction. A slicer software arranges all parts to be printed in a single build job.

**Part packing density on build plate.** Many parts can be placed on a single build plate due to the high packing density of the print process (for example, hundreds of parts in Extended Data Fig. 2). In contrast, powder-based print processes pose thermal constraints that do not allow parts to be placed close to each other. While powder-based systems typically only pack about 15% to 20% (ref. 55), VCJ, as a form of inkjet material deposition, can accommodate packing densities above 40%.

**Print materials.** Three materials were printed together to produce the final parts: soft, rigid and support. A thiol-ene elastomer was used to print soft flexible components (Fig. 2). A rigid formulation of thiol-ene was used as the load-bearing structure. A phase-change material (wax) was used as a support structure. The phase-change material is jetted in a molten state at an elevated temperature and hardens as it cools after deposition. The material melts upon reheating above 60 °C, allowing for easy removal (Extended Data Fig. 2 and Supplementary Video 3). Additionally, VCJ also supports the print of epoxies. Two epoxy formulations[56] have been developed: a tough epoxy (Extended Data Table 1c) and a chemically resistant epoxy (Extended Data Table 1d).

**Multimaterial prints.** Multimaterial fabrication depends on the chemistries in use. In general, multimaterial parts must consist of materials from the same polymer family to ensure adequate bonding when mixed or placed in direct contact with each other. Incompatible materials can refuse to bond, causing separation, or inhibit curing. If multimaterial parts with incompatible materials are needed, it is possible to separate the two material regions with a thin separator of support wax (single voxel) to ensure full cure. This separation benefits from the use of mechanical interlocking between the two material regions to prevent material separation after the support is removed (Extended Data Fig. 1g–i).

## Testing standards and material characterization
We used standardized testing to evaluate the printable materials compared to the state-of-the-art materials. In the following, we describe the standards used in this work.

**Modulus of resilience using ASTM 2632.** We investigated the modulus of resilience of the materials directly from the printer according to ASTM 2632 (ref. 57) with three samples per material. ASTM 2632 specifies the test parameters for impact resilience of solid rubber from the measurement of the vertical rebound of a dropped mass from 16 inches in height.

**Assignment of a glass transition temperature $T_g$ by DMA using ASTM E1640-18.** We conducted the DMA and assigned a glass transition temperature $T_g$ according to ASTM E1640-18 (ref. 58). The DMA was performed on soft thiol-ene (Fig. 2g, Extended Data Fig. 4) and compared with the two acrylates Tango Black Plus and Agilus 30 (Fig. 2g).

**Viscoelastic behaviour.** The viscoelastic behaviour of the materials was quantified by recording stress-strain cycles going from 0% to 140% displacement and back to 0% at a stain rate of about 0.53 s$^{-1}$. The hysteresis of the material that relates to its viscoelasticity can be inferred by the area enclosed by the stress-strain cycle. We tested three samples of soft thiol-ene and Tango Black Plus, and two samples of Agilus 30.

**Outdoor weathering using ASTM G154 Cycle 1.** ASTM G154 (ref. 59) mimics outdoor weathering in addition to UV exposure. The test reproduces the weathering effects that occur when materials are exposed to sunlight and moisture (rain or dew) during real-world usage. Rather than just an exposure to humidity, this test causes water droplets to form on the parts' surface, modelling dew formation.

The testing standard ASTM G154, Cycle 1 exposes all samples to 0.89 W (m$^2$ nm)$^{-1}$ UV irradiation at a wavelength of about 340 nm from a UVA-340 lamp. The exposure cycle consists of 8 h UV at (60 ± 3) °C Black Panel Temperature followed by 4 h Condensation at (50 ± 3) °C Black Panel Temperature. The test samples were removed and tested after 250 h, 500 h, 750 h and 1,000 h.

**Material characterization.** In contrast to processes that require a planarizer, the contactless VCJ process enables printing of chemistries that continue to cure after the discontinuation of irradiation. This includes thiol-ene and epoxy chemistries.

The soft thiol-ene material[60] has a Shore hardness of 32 A, tear resistance of 5.6 kN m$^{-1}$ and elongation at break of 200% (Extended Data Table 1a). In addition, the material's exposure to the outdoors was simulated according to ASTM G154, Cycle 1. After the outdoor weathering, material tests were conducted following ASTM D638: Type IV, 50 mm min$^{-1}$ (Extended Data Fig. 3d–i). Another thiol-ene resin was used to print rigid components. The rigid thiol-ene has a tensile strength of 45 MPa, tensile modulus of 2.1 GPa and elongation at break of 15% (Extended Data Table 1b).

The thiol-ene step-growth polymerization utilized in this work consists of an ABAB system alternating between poly-thiols and poly-enes. This polymerization approach results in a highly regular polymer chain structure, which combined with the high molecular weight, achieved through careful formulation, results in a highly elastic polymer. The high elasticity of the polymer can be seen in the large change of storage modulus before and after the glass transition temperature $T_g$ in the DMA (Extended Data Fig. 4).

The contactless VCJ process also permits the printing of further resin families, for example, 100% UV-cationic cured epoxy materials. Epoxies are particularly attractive for several reasons, including low shrinkage, high chemical resistance and excellent UV stability. The tough epoxy presents an ultimate breakdown strength of 53.8 MPa,

an elastic modulus of 2.5 GPa, an elongation at break of 7.1%, a Shore hardness 78D, Izod impact strength of 33.8 J m$^{-1}$ and a heat deflection temperature at 0.45 MPa of 76 °C (Extended Data Table 1c). In addition, the outdoor stability of the epoxy was tested per ASTM G154, Cycle 1, followed by ASTM D638, Type IV, at 50 mm min$^{-1}$ (Extended Data Fig. 3d).

The chemically resistant epoxy has an ultimate tensile strength of 59.2 MPa, an elastic modulus of 2.7 GPa, an elongation at break of 2.5%, a Shore hardness 81D and a heat deflection temperature at 0.45 MPa of 130 °C (Extended Data Table 1d). This epoxy is also resistant to chemicals and solvents (Extended Data Fig. 3a,c).

The adhesion between cast soft and rigid thiol-ene was tested via lap shear ASTM D 3163-01 (Extended Data Fig. 7). A shear strength of (1.08 ± 0.10) MPa was determined for five tested samples.

### Printed systems and robots

**Robotic hand.** The printed robotic hand resembles a human hand with bones whose shapes have been extracted from open-source magnetic resonance imaging data[37]. The joints connecting the bones are modelled to resemble the human anatomy. The printed tendons are attached to the bones in locations approximating the anatomically correct insertion areas of the muscles. Rigid guides are modelled as extrusion from the bone to guide the tendons to ensure the forces are delivered to the attachment point. Each printed tendon is connected to a servo motor (DYNAMIXEL XL430-W250-T, ROBOTIS Co. Ltd.). One end of multifilament fishing line is knotted to the end of the printed tendon and the other end of the fishing line is spooled onto a reel of the servo motor.

Each fingertip and the palm of the hand are fitted with a sensor pad that measures pressure. This printed sensor pad is a cavity with a thin membrane that is connected through a long, printed tubing. Each printed tubing coming from the sensor pad is externally connected to a commercial pressure sensor (015PG2A3, Honeywell International Inc.) with a sensor range of 0 kPa to 25 kPa. The sensor signal is read out by a microcontroller (Arduino DUE, Arduino S.r.l.).

The hand's controller runs on a computer. The motor's actuation patterns and control sequences are written in Python, and the sensor signal from the microcontroller is read out via a serial connection. The control loop for the hand allows the closing of the individual fingers until contact is sensed through the printed sensor pads.

The hand was evaluated by testing its compliance, dexterity and ability to grasp objects. The fingers' compliance was tested through the manual bending of the joints and hitting the hand with a hammer. The dexterity of the hand was evaluated by controlling the tendon-actuation to make contact between the tip of the thumb and another fingertip of the same hand. The object grasping tests were executed according to a multistep grasping algorithm (Extended Data Fig. 5). Several objects were placed in front of the hand. The closure of the hand was started as soon as contact was sensed at the palm sensor. The fingers then closed until their fingertips sensed contact with the object to be grasped.

**Walking robot.** The printed walking robot prototype is an eight-channel system with two sets of two channels for actuating groups of three legs (Extended Data Fig. 6a). One channel supplies the top joints and one the bottom joints of the group of three legs. Applying pressure to these channels bends the legs at the respective joint. Pressure patterns symmetric to the centre plane of the robot allow the robot to locomote in a forward and backward direction. The pressure patterns are adapted to provide more pressure to one half than the other, leading to the robot turning left or right. Another set of two channels is used to actuate the robot's arm. One actuator is located at the joint intersection with the body. The other actuates the 'forearm'. Finally, two channels connect to a gripper. One channel supplies the gripper with actuation pressure, the other connects the sensing pad to a pressure sensor. The sensing pad is a cavity at the fingertips of the gripper. Reading pressures at these channels allows us to reason about the forces and thereby the contact made between the tip of the gripper and the contacting object.

We connect the supply channels of the robot to seven channels of a 16-channel proportional valve terminal (MPA-FB-VI, Festo Vertrieb GmbH & Co. KG). The valve terminal has individually addressable channels that command pressures between 0 kPa to 250 kPa at a flow rate per channel of up to 380 l min$^{-1}$. The sensing channel is connected to a pressure sensor (015PG2A3, Honeywell International Inc.) with a sensor range of 0 kPa to 25 kPa. A microcontroller (Arduino DUE, Arduino S.r.l.) receives the sensor signal and streams the measurements to the serial port. The pressure patterns and control sequences are written in Python, and the sensor signal from the microcontroller is read from its serial port. We demonstrate the walking robot's ability to locomote, grasp and sense using different objects. Experimental still images (Extended Data Fig. 6b) and video recordings (Supplementary Video 5) are available.

**Heart pump.** The functional heart pump is a multimaterial print that operates as two pressurized-air-driven liquid pumps (Supplementary Video 6) resembling the double ventricle of a mammalian heart. Two openings are located at the bottom of the heart to allow pressurized air to compress the membranes of each artificial ventricle. This compression corresponds to a heart muscle shrinking the volume of a ventricle. The ventricle's volume is connected to a liquid supply system through a one-way inlet valve and a one-way outlet valve. These valves resemble the three-leaved heart valves that can be found in the aortic valve, the tricuspid valve and the pulmonary valve. The outer shell of the heart approximates a mammalian heart. Each ventricular chamber is fitted with a printed sensor pad that allows the sensing of the heart's frequency. The sensor pad connects to a sensor channel in the heart. The channel is connected to a pressure sensor (015PG2A3, Honeywell International Inc.) with a sensor range of 0 kPa to 25 kPa. The sensor signal is decoded on a microcontroller (Arduino DUE, Arduino S.r.l.). A reciprocating syringe pump system is used to actuate the printed pump.

To test the flow rate of the heart and the functionality of the sensor, an experimental setup like the circulatory system found in mammals was used (Supplementary Video 6). Three 10 l translucent buckets were connected to the heart. The left bucket resembled de-oxygenated, old blood, the bucket in the middle resembled the lung's blood volume and the right bucket was for oxygenated blood leaving the heart. To measure the flow rate, we recorded the change in weight of the buckets over time. The sensed frequency in the sensor pads was compared to the frequency of actuation of the syringe pump.

**Multimaterial metamaterial structure.** Going beyond the limited properties of a single material in bulk, metamaterials can be freeform constructed from multiple materials to provide features not found in a homogeneous material block. We can adjust by design the stress-strain curve of a material using a truss-based configuration. The links of the truss are made of soft materials and the nodes of the truss are additionally reinforced with rigid, spherical elements. This configuration allows for more distinct changes in material stiffness beyond a given level of strain.

We printed metamaterials from soft and rigid thiol-ene with different link and node diameters and tested the resulting cubes of the metamaterials using a compression testing machine (Instron 5943, Illinois Tool Works Inc.) and a high-speed camera (FASTCAM Mini AX200, Photron). Each metamaterial construct was placed in the testing area of the compression testing machine and was compressed from 0 mm to 18.2 mm in relative displacement.

### Data availability

All data needed to evaluate the conclusions in the paper are present in the paper and/or the Supplementary Information.

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

**Acknowledgements** We thank Credit Suisse for their generous donation to the ETH Zurich Foundation that established the Soft Robotics Lab by Robert Katzschmann at ETH Zurich. We thank the Swiss National Science Foundation (SNSF) for their support of the Soft Robotics Lab (grant no. 200021_215489). We also thank DARPA (grant no. HR00111970014), the National Science Foundation (grant no. 1722000), Naval Air Warfare Center (grant no. N68335-21-G-005), and Mass Ventures (START Stage I Grant) for their support of this work. We thank R. Kaoutar for her help analysing the metamaterial. We thank J. Marini Parissi for his advice on designing multimaterial objects. We thank R. Woudenberg for his work on the chemistry and process parameters. We thank H. Wang for his work on machine vision. We thank H. Kellogg for his work on the optical scanner of the printer. We thank D. Russell for his work on the 3D motion system. We thank L. DeSimone for his designs that showcase the printer's feature set.

**Author contributions** A contribution to the conception of the work was made by J.R. (printer), D.M. (printer), W.M. (printer and printed systems) and R.K. (printer, characterizations, printed systems and robots). A contribution to the design of the work was made by T.B. (printed systems, robots, and their characterizations), S.R. (walking robot), S.W. (robotic hand), Y.A. (pump), J.R. (printer), A.W. (optical scanner), S.T. (chemistry), G.E. (chemistry), C.K. (jetting hardware), J.J. (printing parameters), W.M. (printer and printed systems) and R.K. (printed systems and their characterizations). A contribution to the creation of new software used in this work was made by T.B. (printed systems and robots), S.R. (robotic walker), S.W. (robotic hand), B.C. (robotic walker and robotic hand), J.R. (printer), D.C. (printer), W.Z. (electronics and firmware), D.K. (electronics and firmware) and R.K. (printed systems and robots). A contribution to the data acquisition was made by T.B. (printed systems, robots, and their characterizations), S.R. (walking robot), S.W. (robotic hand), Y.A. (pump) and J.J. (operating printer and postprocessing). A contribution to the data analysis was made by T.B. (printed systems, robots, and their characterizations), S.R. (walking robot), S.W. (robotic hand), Y.A. (pump), J.R. (printer), A.W. (optical scanner) and R.K. (printed systems, robots, and their characterizations). A contribution to the data interpretation was made by T.B. (printer, printed systems, and robots), S.R. (walking robot), S.W. (robotic hand), Y.A. (pump), J.R. (printer), A.W. (optical scanner) and R.K. (printer, printed systems, and robots). T.B., S.T., G.E., W.M. and R.K. drafted the work. T.B., S.T., W.M. and R.K. substantively revised the work. All authors approved the final draft of the manuscript.

**Funding** Open access funding provided by Swiss Federal Institute of Technology Zurich.

**Competing interests** J.R., W.M. and D.M. are co-founders of Inkbit Corporation. J.R., D.M., A.W., D.C., S.T., G.E., J.J., W.Z., D.K. and C.K. are employees of Inkbit Corporation and own stock in Inkbit Corporation. Inkbit Corporation holds the patents US 11,208,521; US 11,173,667; US 10,456,984 and US 11,155,040 related to print materials and vision-controlled deposition.

**Additional information**
**Correspondence and requests for materials** should be addressed to Wojciech Matusik or Robert K. Katzschmann.

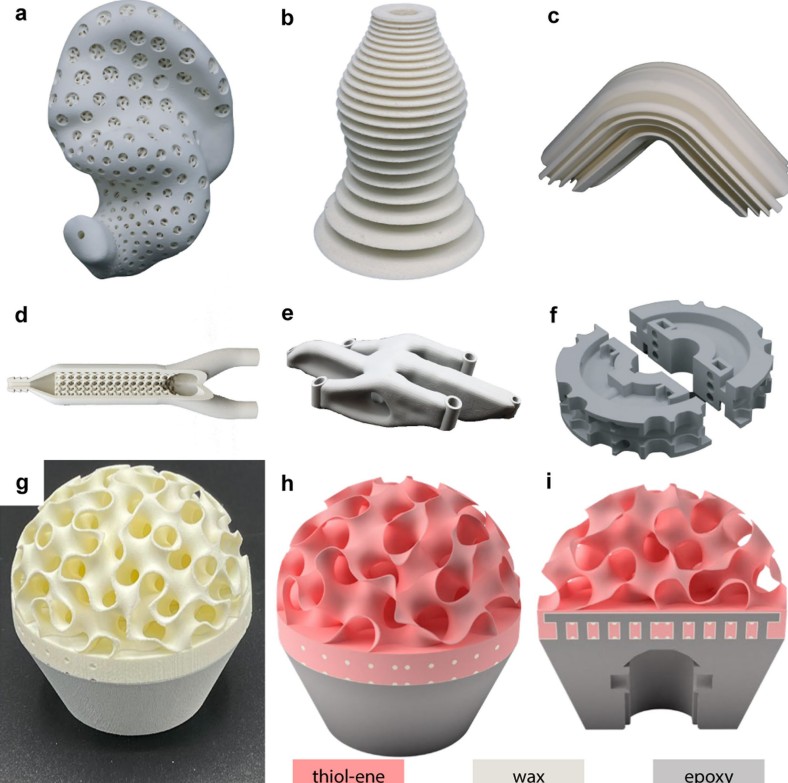

| thiol-ene | wax | epoxy |

**Extended Data Fig. 1 | Illustrating a feature set the printer is capable of printing.** The first row (**a**–**c**) is made from soft thiol-ene, the second row (**d**–**f**) is made from tough epoxy. **a**, This custom earpiece made from soft thiol-ene has a size of 5 cm × 2.5 cm with holes between 0.4 mm to 1 mm. The inner lattice structure's struts are 0.3 mm. **b**, This object made from soft thiol-ene has a size of 5 cm × 2.5 cm the lamella have a wall thickness of 0.9 mm. **c**, This gasket made from soft thiol-ene has a size of 5 cm × 7.5 cm with a wall thickness of 0.75 mm. **d**, This static mixer made from epoxy has a diameter of 5.6 cm at a width of 21.6 cm with an internal strut size of 0.5 mm. **e**, This generative design bracket made from epoxy has a length of 21.6 cm and a width of 11.6 cm. **f**, This generative design bracket made from epoxy has a diameter of 25.4 cm. **g**, Multi-chemistry print of epoxy and thiol-ene mechanically interlocked. The upper (white) half of the prints is made from soft thiol-ene, the lower (grey) half of the print is epoxy-based. **h**, A render of the printed part with colour-coding for thiol-ene in pink, wax in light grey and epoxy in dark grey. **i**, The render of the printed part cut in half where the mechanical interlocking is visible.

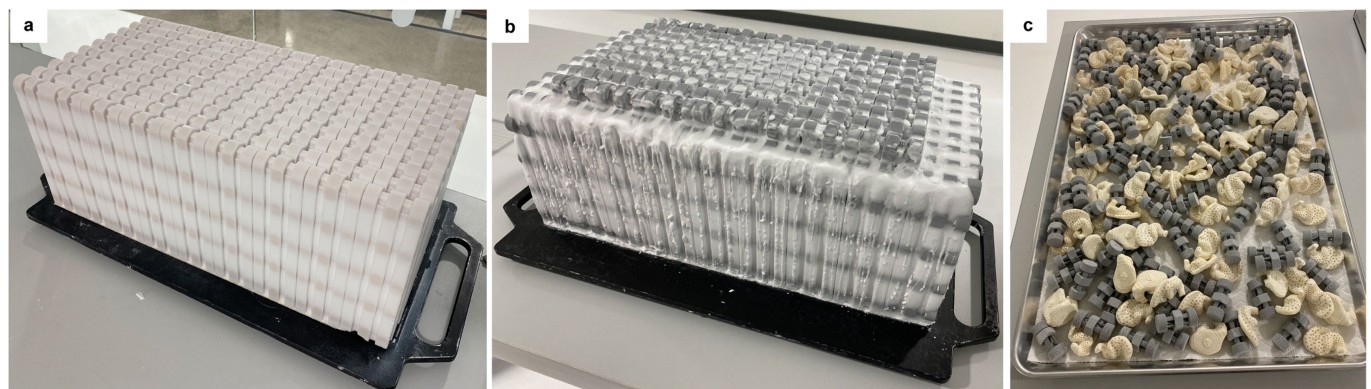

**Extended Data Fig. 2 | The support removal process. a**, The build plate full of printed parts of different material covered in white wax support. The wax support is melted in an oven at 65 °C. **b**, Intermediate image of the support melting process. **c**, After wax melting, the parts are put in an aqueous degreasing solution as the last step of preparation.

## a

| Solvent | Chem. epoxy (5 days) | Tough epoxy (7 days) | RGD 720 (7 days) |
|---|---|---|---|
| Water | 0.3 wt% | 2.7 wt% | 2.5 wt% |
| Acetone | 0.5 wt% | fail | fail |
| Isopropyl alcohol | 0.0 wt% | 4.6 wt% | 3.2 wt% |
| Motor oil | 0.1 wt% | 1.8 wt% | 0.6 wt% |
| Ethanol | 0.5 wt% | 11.5 wt% | 12.9 wt% |

## b

| Solvent | Soft thiol-ene (7 days) | Tango Black Plus (7 days) |
|---|---|---|
| Water | 4.3 wt% | 5.1 wt% |
| Acetone | 89.6 wt% | 139.7 wt% |
| Isopropyl alcohol | 22.9 wt% | 106.4 wt% |
| Motor oil | 0.9 wt% | 5.1 wt% |
| Ethanol | 39.5 wt% | fail |

## c

| Solvent | Tensile strength Abs. (MPa) | Tensile strength Change w.r.t. control | Tensile modulus Abs. (GPa) | Tensile modulus Change w.r.t. control | Elongation at break Abs. (%) | Elongation at break Change w.r.t. control |
|---|---|---|---|---|---|---|
| Control | 53.0 | 0.0 % | 2.70 | 0.0 % | 2.5 | 0 % |
| Nitric acid (2%aq) | 51.0 | -3.8 % | 2.60 | -3.7 % | 2.6 | 0 % |
| Sulfuric acid (20%aq) | 53.3 | 0.6 % | 2.60 | -3.7 % | 2.6 | 4 % |
| Acetone | 69.0 | 30.2 % | 2.60 | -3.7 % | 7.6 | 204 % |
| Ethanol | 66.8 | 26.1 % | 2.70 | 0.0 % | 4.0 | 60 % |
| NaOH (20%w/w) | 51.3 | -3.2 % | 2.60 | -3.7 % | 2.4 | -4 % |
| Isopropyl alcohol | 52.6 | -0.8 % | 2.73 | 1.1 % | 2.4 | -4 % |
| Tolulene | 57.8 | 9.1 % | 2.67 | -1.1 % | 2.8 | 12 % |
| Bleach | 52.5 | -0.9 % | 2.79 | 3.3 % | 2.5 | 0 % |
| Synthetic sea water | 51.4 | -3.0 % | 2.74 | 1.5 % | 2.3 | -8 % |
| Motor oil | 54.0 | -1.9 % | 2.77 | 2.6 % | 2.4 | -4 % |

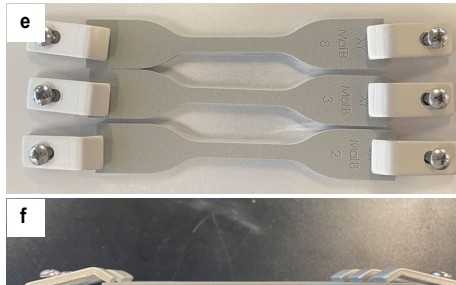

## d

| Exposure (h) | Tensile strength Absolute (MPa) | Tensile strength Change w.r.t. control | Tensile modulus Absolute (GPa) | Tensile modulus Change w.r.t. control | Elongation at break Absolute (%) | Elongation at break Change w.r.t. control |
|---|---|---|---|---|---|---|
| 0 | 53.8 | 0 % | 2.50 | 0 % | 7.1 | 0 % |
| 250 | 54.9 | 2 % | 2.42 | -3 % | 8.8 | 24 % |
| 500 | 58.1 | 8 % | 2.43 | -3 % | 6.7 | -6 % |
| 750 | 59.2 | 10 % | 2.40 | -4 % | 6.4 | -10 % |
| 1000 | 57.0 | 6 % | 2.40 | -4 % | 6.6 | -7 % |

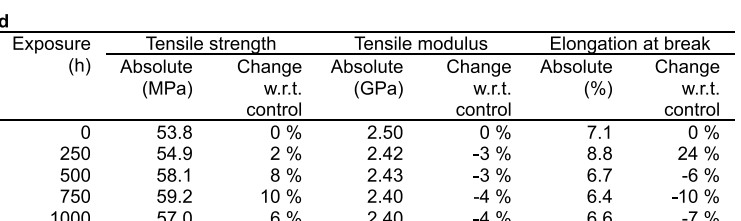

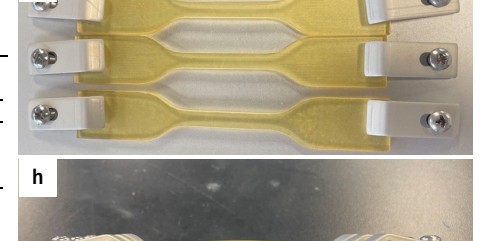

## i

| Exposure (h) | Tensile strength Absolute (MPa) | Tensile strength Change w.r.t. control | Tensile elongation Absolute (%) | Tensile elongation Change w.r.t. control | Elastic modulus Absolute (MPa) | Elastic modulus Change w.r.t. control | Tear strength Absolute (kN/m) | Tear strength Change w.r.t. control | Tear propagation Absolute (kN/m) | Tear propagation Change w.r.t. control |
|---|---|---|---|---|---|---|---|---|---|---|
| 0 | 0.92 | 0 % | 200 | 0 % | 0.53 | 0 % | 3.7 | 0 % | 5.6 | 0 % |
| 250 | 0.81 | -12 % | 200 | 0 % | 0.48 | -9 % | 3.4 | -8 % | 5.7 | 2 % |
| 500 | 0.79 | -14 % | 190 | -5 % | 0.50 | -6 % | 3.4 | -8 % | 6.2 | 11 % |
| 750 | 0.86 | -6 % | 230 | 15 % | 0.47 | -11 % | 3.7 | -0 % | 7.0 | 25 % |
| 1000 | 0.70 | -14 % | 190 | -5 % | 0.51 | -4 % | 3.5 | -5 % | 6.3 | 13 % |

**Extended Data Fig. 3 | Exposure tests for printed materials. a**, Weight gain for rigid polymers under solvent exposure according to ASTM D543. **b**, Weight gain for soft polymers under solvent exposure according to ASTM D543. **c**, Change in material properties for chemically resistant epoxy after five-day submersion at 25 °C, following ASTM D543. Tests conducted following ASTM D638: Type IV, 50 mm/min, average values represented. **d**, Tough epoxy accelerated aging test per ASTM G154, Cycle 1 over 1000 h exposure, average values represented. Top view (**e**, and **g**) and side view (**f**, and **h**) images of material samples in the test setup for exposure according to ASTM G154, Cycle 1. Our tough epoxy material (**e**, and **f**) did not show warping compared to the state-of-the-art acrylate-based material (Stratasys RGD720) (**g**, and **h**). **i**, Soft thiol-ene accelerated aging test ASTM G154, Cycle 1 over 1000 h exposure. Tests conducted following ASTM D412: Die C, 500 mm/min, ASTM D624-C, and ASTM D624-B, average values represented.

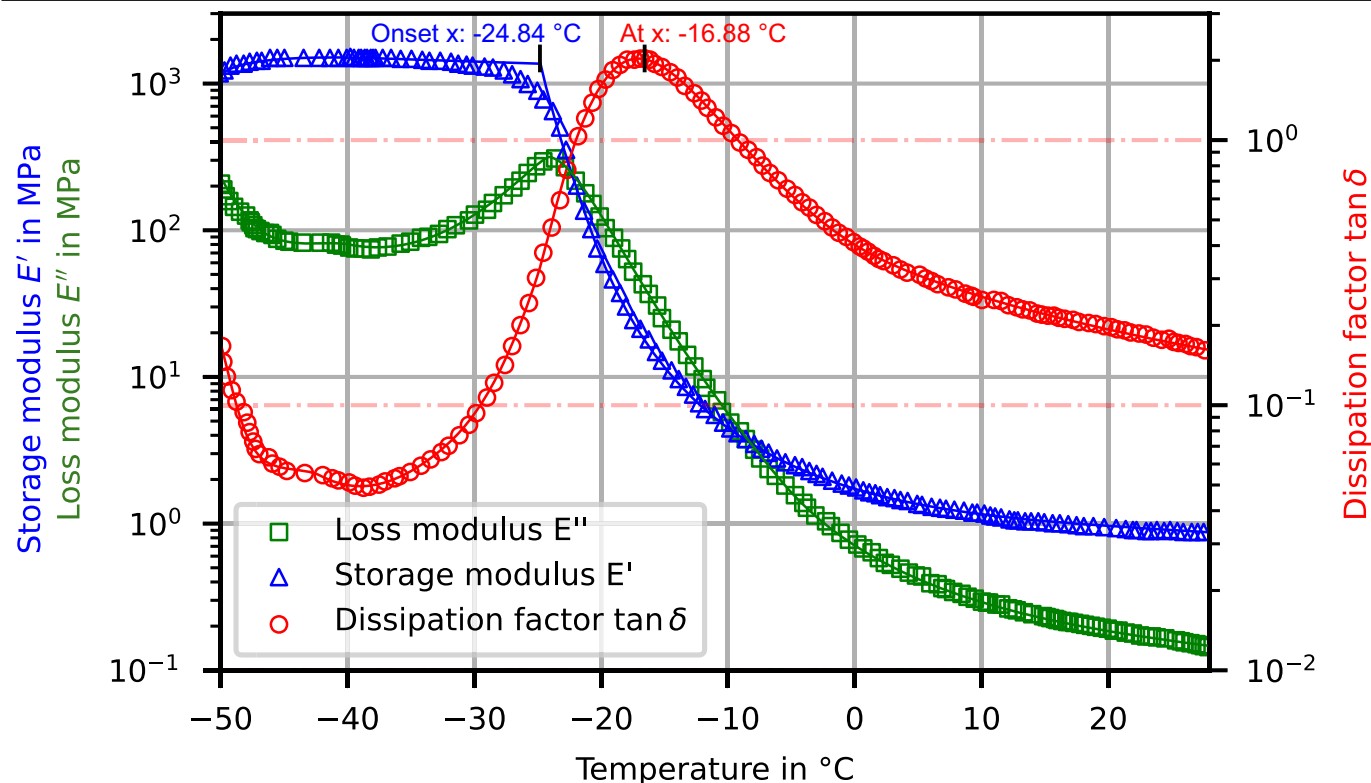

**Extended Data Fig. 4 | Soft thiol-ene dynamic mechanical analysis.** The onset of the drop in storage modulus E′ happens at −24.84 °C. The dissipation factor tan(δ) peaks at −16.88 °C. This measurement was taken at another time than DMA in Fig. 2g.

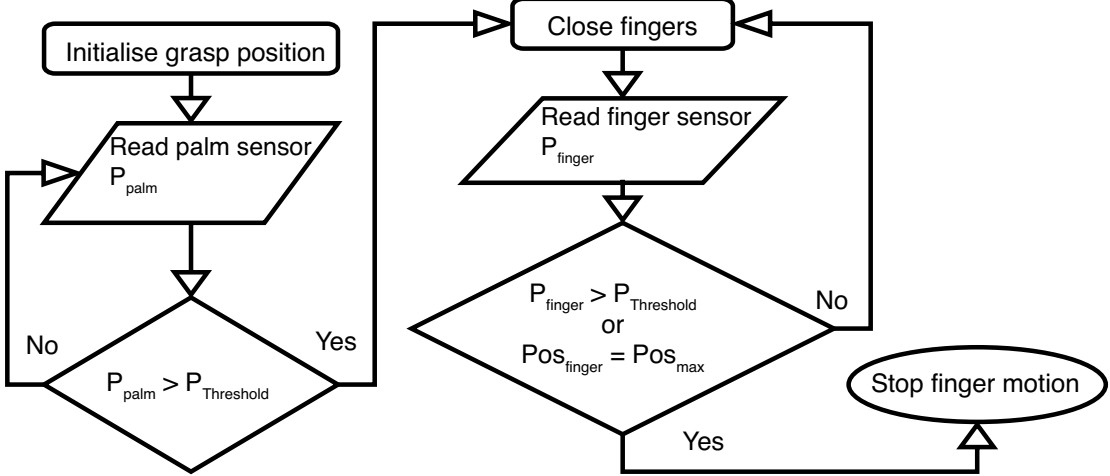

**Extended Data Fig. 5 | Control diagram for a hand grasp.** Illustration of the control algorithm used for the grasps shown in the results.

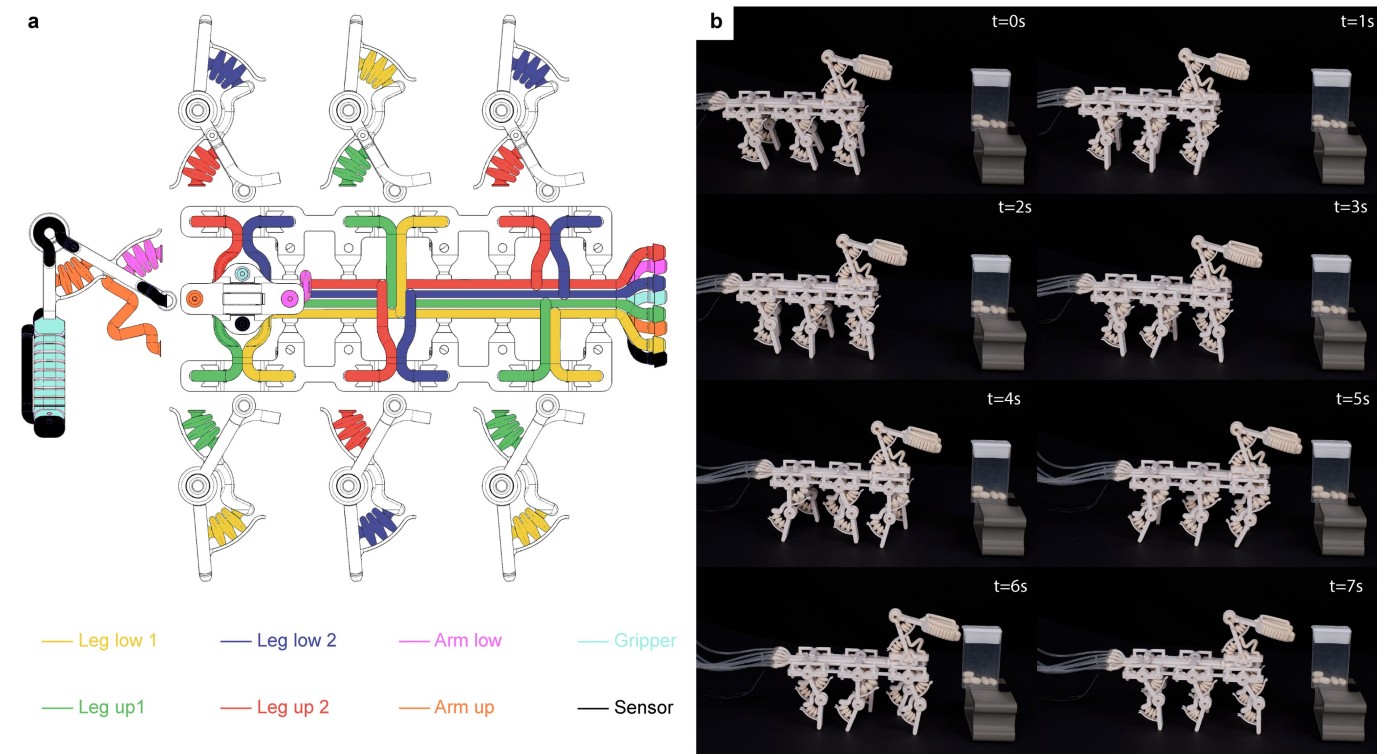

**a**

— Leg low 1 — Leg low 2 — Arm low — Gripper

— Leg up1 — Leg up 2 — Arm up — Sensor

**Extended Data Fig. 6 | The walking robot's channels and actuation. a**, Visualization of the walking robot's channels and respective actuators and sensor pads are colour coded. **b**, Sequential still images are taken every second during the locomotion of the walking robot.

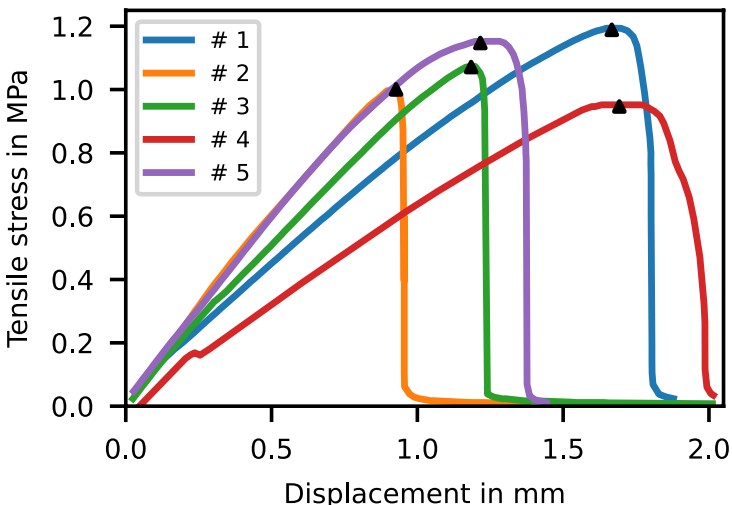

**Extended Data Fig. 7 | Lap shear testing for cast soft and rigid thiol-ene.** A shear strength of (1.08 ± 0.10) MPa was determined by testing five samples in accordance with ASTM D 3163-01.

## Extended Data Table 1 | Properties for printed materials

**a**

| Mechanical properties | Standard | Value |
|---|---|---|
| Ultimate tensile strength | ASTM D412-C | 0.92 MPa |
| Elongation at break | ASTM D412-C | 200 % |
| Elastic modulus @ 100 % | ASTM D412-C | 0.53 MPa |
| Tear propagation | ASTM D624-B | 5.6 kN/m |
| Tear strength | ASTM D624-C | 3.7 kN/m |
| Shore hardness @ 100 % | ASTM D2240 | 32 A |
| Compression set, 23 °C, 72 h | ASTM D395-B | 10.2 % |
| **Thermal properties** | | |
| Glass transition temperature | ASTM D3418 | -32 °C |

**b**

| Mechanical properties | Standard | Value |
|---|---|---|
| Ultimate tensile strength | ASTM D638-IV | 45 MPa |
| Elongation at break | ASTM D638-IV | 14.6 % |
| Elastic modulus @ 100 % | ASTM D638-IV | 2.1 GPa |
| Shore hardness | ASTM D2240 | 81 D |
| **Thermal properties** | | |
| HDT @ 0.45 MPa | ASTM D648 | 52 °C |
| HDT @ 0.45 MPa | ASTM D648 | 48 °C |
| Glass transition temperature | ASTM D3418 | 66 °C |

**c**

| Mechanical properties | Standard | Value |
|---|---|---|
| Ultimate tensile strength | ASTM D638-IV | 53.8 MPa |
| Elongation at break | ASTM D638-IV | 7.1 % |
| Elastic modulus | ASTM D638-IV | 2.5 GPa |
| Flexural strength | ASTM D790 | 85 MPa |
| Flexural modulus | ASTM D790 | 2.3 GPa |
| Shore hardness | ASTM D2240 | 78 D |
| Izod impact (notched) | ASTM D256 | 33.8 J/m |
| **Thermal properties** | | |
| HDT @ 0.45 MPa | ASTM D648 | 76 °C |
| HDT @ 0.45 MPa | ASTM D648 | 69 °C |
| Glass transition temperature | ASTM D3418 | (78 to 82) °C |
| **General properties** | | |
| Density | ASTM D792 | 1.21 g/cm$^3$ |
| Water absorption | ASTM D570 | 0.67 % |
| Flammability | UL 94 | HB (3.3 mm) |

**d**

| Mechanical properties | Standard | Value |
|---|---|---|
| Ultimate tensile strength | ASTM D638-IV | 59.2 MPa |
| Elongation at break | ASTM D638-IV | 2.5 % |
| Elastic modulus | ASTM D638-IV | 2.7 GPa |
| Flexural strength | ASTM D790 | 71.7 MPa |
| Flexural modulus | ASTM D790 | 2.4 GPa |
| Shore hardness | ASTM D2240 | 81 D |
| **Thermal properties** | | |
| HDT @ 0.45 MPa | ASTM D648 | 130 °C |
| HDT @ 0.45 MPa | ASTM D648 | 116 °C |
| Glass transition Temperature | ASTM D3418 | 131 °C |
| **General properties** | | |
| Density | ASTM D792 | 1.21 g/cm$^3$ |
| Cytotoxicity | ISO 10993-5 | Pass (24h) |

Material properties for **a**, soft thiol-ene, **b**, rigid thiol-ene, **c**, tough epoxy, and **d**, chemically resistant epoxy.

## Extended Data Table 2 | Print parameters

**a**

| Relative deviation | Linear accuracy Nominal length | | Circular accuracy Nominal diameter | | | | |
|---|---|---|---|---|---|---|---|
| | 0 mm to 25 mm | 25 mm to 55 mm | 7 mm | 8 mm | 15 mm | 23.5 mm | 25 mm |
| X & Y | ±0.1 % | ±0.1 % | -0.03 % | 0.10 % | 0.01 % | 0.13 % | 0.11 % |
| Z | ±0.2 % | ±0.3 % | -0.10 % | 0.14 % | -0.10 % | 0.07 % | 0.02 % |

**b**

| | Pin (mm) | Hole (mm) | Slot/gap (mm) | Rib/wall (mm) | | Unsupported wall (mm) | |
|---|---|---|---|---|---|---|---|
| | | | | Resolved | Undeformed | Resolved | Undeformed |
| Tough epoxy | 0.3 | 0.4 | 0.40 | 0.1 | 0.6 | 0.2 | > 1.0 |
| Chem. epoxy | 0.3 | 0.3 | 0.06 | 0.6 | 0.6 | 0.4 | 0.6 |
| Soft thiol-ene | 0.3 | 0.3 | 0.08 | 0.2 | 0.6 | 0.1 | > 1.0 |

**c**

| | Embossed text (mm) | | Debossed text (mm) | |
|---|---|---|---|---|
| | Height | Depth | Height | Depth |
| Tough epoxy | 1.4 | 0.1 | 2.0 | 0.3 |
| Chem. epoxy | 1.4 | 0.2 | 1.6 | 0.2 |
| Soft thiol-ene | 1.4 | 0.1 | 1.0 | 0.2 |

**d**

| Fitment diameter (mm) | 3D channel diameter (mm) | Drainage hole (mm) | |
|---|---|---|---|
| | | Recommended | Possible |
| 0.1 to 0.2 | 0.1 to 0.2 | ≥ 0.75 | ≥ 0.4 |

**a**, Accuracy for prints with tough epoxy. – ISO/ASTM 52902 **b**, Resolution for prints (in X, Y, and Z). – ISO/ASTM 52902 **c**, Bossed feature size. **d**, Design considerations.