## [Peer Review File · Nature]

Manuscript Title: Vision-Controlled Jetting for Composite Systems and Robots

Reviewer Comments & Author Rebuttals

Reviewer Reports on the Initial Version:

Referees' comments:

Referee #1 (Remarks to the Author):

In their manuscript "Vision-Controller Jetting for Composite Systems and Robotics", Buchner et al. present a new method of ensuring planarization between successive layers in an inkjet/material jetting process. Notably, existing material jetting methods (such as the Polyjet system developed by Stratsys Inc.) rely on the use of a roller and blade to planarize between successive layers to prevent a build-up of height imperfections. This planarization step is a notorious point of failure for material jetting systems, with regular maintenance required on the roller and blade systems to prevent material build-up. Here, the authors propose the use of a laser profilometer to generate a high-resolution topographical map of the build surface, and print control software that steers deposition to compensate for non-planar surfaces. The printer is then used to generate high accuracy and multi-material robotic components such as a hand that can grip and sense, a walker, a metamaterial lattice and a valved heart pump.

Overall, while the high quality of the printed constructs is notable, there is insufficient material characterization to demonstrate an impactful advance in material jetting 3D printing. Notably, vision-controlled material jetting that compensates for non-planar errors has already been published (albeit with low speed), as noted by the authors [DOI: 10.1145/2766962]. Furthermore, the printer resolution here (32 μm x 64 μm x 20 μm) is comparable to that of a Stratasys Polyjet system (typically ~ 43 μm x 43 μm x 16 μm). The existing Polyjet or similar multimaterial inkjet systems have previously been used to print compliant multi-material walkers [10.1109/ICRA.2016.7487576], grippers [DOI: 10.25046/aj040204], pumps [10.1109/ICRA.2016.7487576], and metamaterial lattices [DOI: 10.1126/sciadv.1602890].

Thus, as noted by the authors, the question of whether this represents a fundamental and highly significant advance lies in whether the thiol-ene materials used here offer significant (order of magnitude effect) property enhancements. Despite material jetting's unmatched ability to generate multi-material matter, it is widely considered that the Achilles heel of all material jetting systems lies in the poor material properties of printed components. This shortcoming is largely due not to the photochemistry but to the physics of droplet formation that requires the use of highly non-viscous resins. This viscosity limitation precludes the use of long-chain oligomers that are typically required to achieve high toughness and fatigue resistance. Other processes like DIW and stereolithography can operate with far more viscous resins, and thus typically produce parts with superior material properties. Thus, components produced by material jetting systems like the Polyjet are typically weak, brittle, and highly prone to fatigue. However, from the material data provided, the material properties do not seem significantly better than those obtainable from existing acrylate-based material jetting resins. Specifically, the existing Polyjet process can multi-material print highly rigid (RGD 720; Tensile Strength 50-65 MPa, Elongation at failure 10-25% [1]) and highly flexible materials (Tango Black Plus; Tensile Strength 0.8-1.5 MPa, Elongation at failure 170-220% [1]). In comparison the materials described here have comparable properties (Rigid Thiol-ene; Tensile Strength 45 MPa, Elongation at Failure 14.6%), (Soft Thiol-ene; Tensile Strength 0.92 MPa, Elongation at failure 200%).

I believe side-by-side cyclic tests of 'just printed' and 'pre-aged' materials would need to be performed comparing the fatigue performance of jetted acrylates and thiol-enes (i.e. Tango Black

Plus and the Thiol-ene Soft). While good UV stability of thiol-enes is demonstrated, this should also be a side-by-side comparison of an existing equivalent acrylate material. Also, toughness tests, such as notched Izod, should be performed to compare the performance of material jetted acrylate and thiol-enes. If these tests were to demonstrate order-of-magnitude improvement in properties, then the impact of this work would be much more apparent. Absent more robust material property characterization, I do not have the necessary data to say whether this manuscript represents a fundamental improvement over the status quo in material jetting systems.

Minor Comments:

Purely elastic soft polymers...with similar properties to those of muscles, tendons, and other complex soft tissue...". Soft tissues, such as skeletal muscle, are viscoelastic - not purely elastic.

"Soft polymers can be smeared and squished by a roller" – how this is a disadvantage should be clarified. I.e. loss of resolution and spatially varying or mixing of material composition.

"...Builds the polymer's structure with more precision" – this wording is unclear. A more precise discussion of step-growth benefits on degree of conversion and control over chain length and crosslinking density should be provided".

[1] - Material Data Sheets for Tango Black Plus and RGD 720 accessed from stratasys.com

Referee #2 (Remarks to the Author):

A. Summary of the key results:

The manuscript describes a vision-based inkjet printing process, coined as "Vision-controlled jetting (VCJ)" that achieves a close-loop feedback optimization of the print layer height by capturing the topological information during printing. The contactless vision-based system removes the need for a mechanical planarizer, which expands the possible materials families that can be printed by allowing continuous curing chemistries. The authors demonstrate impressive multimaterial printing capability with several examples of multi-material actuators and architecture that are capable of integrating both rigid and soft materials.

Overall, the manuscript describes an inkjet printing platform capable of creating complex and integrated multi-material (e.g., soft and rigid polymer) constructs. However, the manuscript's scientific novelty and advancement over previous work and published public information are not clear. The manuscript also lacks the quantitative study and data needed to evaluate the key scientific claim (e.g., the quantitative data demonstrating the advancement of its closed-loop optimization in comparison to the earlier work, and mechanical properties of the multi-material structure). There are no discussions of error, potential limitations of the materials, mechanical characterization of the multi-material interface, etc as discussed below:

B: Originality and significance:

In regards to originality:

It is unclear what the specific conceptual novelty and originality of the proposed work are when compared to previous publications, such as published patents. For example, US10456984B2 (citation 59, 60) already describes the similar usage of "closed-loop adaptive material deposition apparatus and method uses a scanning system". The authors did cite these patents when describing the method under "Print control software", but did not discuss and compare the novelty and originality of their work in comparison with these advancements.

Similarly, the idea of machine vision-assisted printing correction is not original, It was mentioned by the authors but the discussion is not well quantified. For instance, the authors claim that the previous work by Sitthi-Amorn, P. et al [ref 56] is "too slow"; but did not provide sufficient reasoning on "why & how" and the quantitative data that compares the relative difference achieved by the current work and the previous work.

It also appears that most of the novelty presented here has already been commercialized and published on some of the authors' company websites (inkbit3d.com), which was not explicitly referenced and highlighted. In fact, some of the videos on the website (<https://inkbit3d.com/technology/>) are identical to the videos that have been submitted for consideration in this manuscript. It is unclear what is the original and novel conceptual and scientific advancement that is being discussed in this manuscript in comparison to public information, especially when many of the claims (e.g., multimaterial adhesion strength, materials compatibilities) are not substantiated with experimental characterizations, but are limited to visual proof of concept.

Broadness of claim and significance:

While the proposed approach appears to be a significant advancement of the current multijet printing approach, some of the claims are not specific and stretch too far beyond. For instance, the claim that the first step of the "organism that mimics nature", as the demonstrated example remains limited to a few selected materials compatible with the inkjet printing process.

C: Data & methodology: validity of approach, quality of data, quality of presentation

The authors did not adequately relate or perform the data and characterization that are critical to understanding the core proposed novelty, especially when comparing the advancement of this work with the earlier work.

For instance, the accuracy of the scanning, and the accuracy of the print compensation are not discussed in the main text. The Tables in supporting information could have been better described and related to the main text.

Further, a core part of the manuscript's novelty is its multi-material capability: but the critical information such as the interfacial strength, and fatigue strength of the multi-material interface are not discussed, which limits the understanding of the application of the described approach.

D: Appropriate use of statistics and treatment of uncertainties

There is insufficient statistical information provided in some of the data (e.g., Tables in supporting information) to evaluate the uncertainty and robustness of the presented data. There is no discussion on the sample size, the standard deviation of the data, etc.

E: Conclusions: robustness, validity, reliability

Additional experimental data, analysis, and discussion can help quantitatively evaluate the robustness, validity, and reliability of the robustness of several key results, such as the strength of the multimaterial structure; particularly when comparing the attributes with the earlier approach.

F: Suggested improvements: experiments, data for possible revision

Several suggestions, as described above, are provided below:

- The novelty of the proposed work can be quantitatively compared with the earlier work and

published information. E.g., when discussing another approach as “too slow”, what is the relative speed improvement that has been achieved, and how?

- The novelty of the work can be more clear if it is focused on what are the specific contributions of the work – e.g., if the advancement is on the scanning strategy – what are the scientific advancements here that achieve what was not possible?
- There is an inadequate discussion on what the limitation of the material is with the proposed approach, and how that impacts its potential application.
- The claim of the broadness of impact can be more specific and relevant to the demonstrated data.
- Critical data describing the fundamental attribute can be made the focus of the work instead of extensive figures that focuses primarily on different applications. While the applications are impressive - it does not adequately provide the quantitative information needed to understand the strength, uniqueness, and limitation of the approach.

G: References: appropriate credit to previous work?:

Yes, the majority of the reference is appropriate

However, some citations are not discussed in the introduction, and some of the submitted materials are publicly available material (published on the author's company website) but are not referenced in the manuscript. It is unclear which content is made original to the manuscript, and which content is a reproduction of the authors' published earlier work.

H: Clarity and context: lucidity of abstract/summary, appropriateness of abstract, introduction and conclusions:

The abstract, introduction, and conclusion can be revised to be more specific to the advancement that is made. The claim that this is the “first step to quickly and reproducibly create organisms that mimic nature” seems too far-fetched, especially when the material limitation here is not adequately described.

Further, it is not clear whether the highlighted attribute, such as the jetting speed and the number of nozzles is the unique core novelty of the work. If the jetting speed and the number of nozzles have already been achieved by the previous inkjet printing approach, highlighting these as a key attribute can cause confusion to the general reader who may not be familiar with inkjet printing on what are the key advancement of the work.

The authors have included many beautifully crafted figures of the 3D models and impressive demonstrations with the 3D printed prototypes. However, many of the presentations of the figures, especially those that involve quantitative data can be improved. Many of the fonts are too small and difficult to read, and some of the labels overlap with the grid line.

Minor comments: There are a few typos: for example pg 27 – the first quotation mark for “forearm” is inverted.

Author Rebuttals to Initial Comments:

Response to Referees of the manuscript titled:

“Vision-Controlled Jetting (VCJ) for Composite Systems and Robots”

Authors: Thomas J. K. Buchner, Simon Rogler, Stefan Weirich, Yannick Armati, Barnabas Gavin Cangan, Javier Ramos, Davide Marini, Aaron Weber, Desai Chen, Scott Twiddy, Greg Ellson, Joshua Jacob, Walter Zengerle, Dmitriy Katalichenko, Chetan Keny, Wojciech Matusik, and Robert K. Katzschmann

Overview of this Response Letter:

Response to Reviewer #1 Page 1ff

Response to Reviewer #2 Page 11ff

Response to Reviewer #1

In their manuscript “Vision-Controller Jetting for Composite Systems and Robotics”, Buchner et al. present a new method of ensuring planarization between successive layers in an inkjet/material jetting process. Notably, existing material jetting methods (such as the Polyjet system developed by Stratsys Inc.) rely on the use of a roller and blade to planarize between successive layers to prevent a build-up of height imperfections. This planarization step is a notorious point of failure for material jetting systems, with regular maintenance required on the roller and blade systems to prevent material build-up. Here, the authors propose the use of a laser profilometer to generate a high-resolution topographical map of the build surface, and print control software that steers deposition to compensate for non-planar surfaces. The printer is then used to generate high accuracy and multi-material robotic components such as a hand that can grip and sense, a walker, a metamaterial lattice and a valved heart pump.

We like to thank the reviewer for many insightful and valuable remarks, which have enabled us to further improve the manuscript. We are glad that the reviewer agrees that we are tackling a notoriously challenging problem with material jetting systems and that we are solving it with a scalable, contactless solution. We would like to highlight that this work not only enables contactless printing, but also shows how new printing chemistries such as thiol-enes and epoxies are now possible. In the following, we address all the reviewer’s remarks.

Remark 1A

Overall, while the high quality of the printed constructs is notable, there is insufficient material characterization to demonstrate an impactful advance in material jetting 3D printing.

We agree that adding detailed material characterization further strengthens the study and better highlights the impact of the work. We have added extensive amounts of data on material characterization through an additional figure and text in the main manuscript. The data shows an order of magnitude improvement in material parameters. The following responses to the reviewer’s remarks provide details on additions made.

Remark 1B

Notably, vision-controlled material jetting that compensates for non-planar errors has already been published (albeit with low speed), as noted by the authors [DOI: 10.1145/2766962].

Previous work from Sitthi-Amorn et al. 2015 [DOI: 10.1145/2766962] introduced a method based on optical coherence tomography scanning. The process was very slow (660 times slower) and limited to simple acrylate resins, simply inadequate for advanced functional systems. We therefore believe that the work contributed here with its new printer design, scanning system, material chemistry, and functional robotic demonstrations provides a substantial and unique advance in research. Multimaterial printed real-world systems that show complex functions are now actually feasible because of the newly allowable chemistries.

We notice that this detail was not properly explained in the previous manuscript and therefore have made the following additions and changes to the revised manuscript:

Line 198ff

Previous work called MultiFab [43] relied on slow full-field optical coherence tomography (OCT) scanning of a small area of only 2 cm × 2 cm. To get information on the surface structure of the full print surface, it had to be rasterised and each area had to be scanned. Next, all scanned areas had to be stitched together in post-processing. Only then could the corrections for the next layer be computed. This post-processing leads to a limited throughput of only 0.05mLmin⁻¹ and print times per layer on the order of minutes, even for smaller prints. MultiFab therefore cannot keep up with inkjet deposition speeds and does not scale to produce large functional parts such as the ones presented in this work. Furthermore, previous work [43] only supported simple acrylate resins, which are inadequate for advanced functional systems.

Line 256ff

Our method utilizes a high-speed 3D vision system to capture a depth map of the currently printed surface, and it compensates for deviations from an ideal planar surface by locally adapting the amount of material to be jetted in the next layer. The method's vision system uses four cameras and two laser sources for laser profilometry while printing. The feedback-loop including the surface scan of the whole print area are performed at printing speed and therefore do not slow down the print process. This leads to a throughput of up to 33mLmin⁻¹, which is 660 times faster than the previous work [43].

Line 984ff

The integrated systems and robots presented in this work were all 3D printed using a multimaterial additive manufacturing platform that utilised a vision-controlled jetting technology. ~~Previous work [43] relied on slow optical coherence tomography (OCT) scanning, which cannot keep up with inkjet deposition speeds and does not scale to produce large functional parts such as the ones presented in this work. Furthermore, previous work [43] only supported simple acrylate resins, which are inadequate for advanced functional systems.~~

Remark 1C

Furthermore, the printer resolution here (32 μm × 64 μm × 20 μm) is comparable to that of a Stratasys Polyjet system (typically ~43 μm × 43 μm × 16 μm).

We agree with the reviewer that we did not attribute the novelty of the work well. The print head, its resolution, and the print speed are not a contribution of this work. We adapted the manuscript to clarify this better:

Line 282ff

The printer's build volume is 500mm × 245mm × 200 mm. The print heads and the print speed are therefore on-par with state-of-the-art inkjet printers for this scale and resolution but our print system including the scanner allows for the first time a wider range of chemistries and therefore material properties compared to other printers.

Remark 1D

The existing Polyjet or similar multimaterial inkjet systems have previously been used to print compliant multi-material walkers [10.1109/ICRA.2016.7487576], grippers [DOI: 10.25046/aj040204], pumps [10.1109/ICRA.2016.7487576], and metamaterial lattices [DOI: 10.1126/sciadv.1602890].

Thank you for making this comment on these three research papers on printing. Please let us address each of them one-by-one in the following:

A dual material walker (rigid and liquid), gripper (soft and liquid), and pump (rigid and liquid) [10.1109/ICRA.2016.7487576] (MacCurdy et al., Printable hydraulics): These demonstrations based on acrylates are not in the range of elasticity and viscosity required for functional systems that last longer than a few actuations. Work only showed single solid material plus uncured liquid and a hard-to-remove gel-like support. Besides support material, this work did not print two solids of different softness, but just one solid structural material. This work was already cited in the manuscript.

An assembled gripper [DOI: 10.25046/aj040204] (Dämmer et al., Design of an Additively Manufacturable Multi-Material Light-Weight Gripper): Additively designed and printed but still much assembly work of individually printed units necessary. This work is now also cited in the manuscript.

Metamaterial lattices [DOI: 10.1126/sciadv.1602890] (Ding et al., Direct 4D printing via active composite materials): These were printed using an existing Polyjet system with acrylates TangoBlack and VeroClear. Results in a bi-material flat design made of shape memory polymers where thermal changes deform a two-dimensional surface.

All the works are now cited in the manuscript:

Line 152ff

Printed multimaterial bellows can assembled to a suction gripper [37]. Different inks can be interspersed to create discrete changes in the material stiffness [38]. Soft and rigid acrylates can also be used as layered shape memory polymers in thin structures [39]. In addition to acrylate resins, liquids that do not cure can be jetted to produce hydraulic systems [40, 41].

References:

[37] Dämmer, G., Gablenz, S., Hildebrandt, A. & Major, Z. Design of an Additively Manufacturable Multi-Material Light-Weight Gripper with integrated Bellows Actuators. *Advances in Science, Technology and Engineering Systems Journal* 4 (2019). <https://doi.org/10.25046/aj040204> .

[38] Bartlett, N. W. et al. A 3D-printed, functionally graded soft robot powered by combustion. *Science* 349 (6244), 161–165 (2015). <https://doi.org/10.1126/science.aab0129> .

[39] Ding, Z. et al. Direct 4D printing via active composite materials. *Science Advances* 3 (4), e1602890 (2017). URL <https://doi.org/10.1126/sciadv.1602890> .

[40] MacCurdy, R., Katzschmann, R., Kim, Y. & Rus, D. Okamura, A. (ed.) Printable hydraulics: A method for fabricating robots by 3D co-printing solids and liquids. (ed.Okamura, A.) 2016 IEEE International Conference on Robotics and Automation (ICRA), 3878–3885 (2016). URL <https://doi.org/10.1109/ICRA.2016.7487576>.

Remark 1E

Thus, as noted by the authors, the question of whether this represents a fundamental and highly significant advance lies in whether the thiol-ene materials used here offer significant (order of magnitude effect) property enhancements.

Thank you for this remark. Now we provide ample results in the section on thiol-ene chemistry. We explain how a difference in the polymerization chemistry leads to better control and enhancement of material properties. This is especially shown in regard to an order of magnitude improvements when tested for UV stability and a substantial improvement in its operational temperature range. We provide a detailed list of changes in the answer to **remark 1F**.

Remark 1F

Despite material jetting's unmatched ability to generate multi-material matter, it is widely considered that the Achilles heel of all material jetting systems lies in the poor material properties of printed components. This shortcoming is largely due not to the photochemistry but to the physics of droplet formation that requires the use of highly non-viscous resins. This viscosity limitation precludes the use of long-chain oligomers that are typically required to achieve high toughness and fatigue resistance. Other processes like DIW and stereolithography can operate with far more viscous resins, and thus typically produce parts with superior material properties. Thus, components produced by material jetting systems like the Polyjet are typically weak, brittle, and highly prone to fatigue.

*However, from the material data provided, the material properties do not seem significantly better than those obtainable from existing acrylate-based material jetting resins. Specifically, the existing Polyjet process can multi-material print highly rigid (**RGD 720**; Tensile Strength 50-65 MPa, Elongation at failure 10-25% [1]) and highly flexible materials (**Tango Black Plus**; Tensile Strength 0.8-1.5 MPa, Elongation at failure 170-220% [1]). In comparison the materials described here have comparable properties (Rigid Thiol-ene; Tensile Strength 45 MPa, Elongation at Failure 14.6%), (Soft Thiol-ene; Tensile Strength 0.92 MPa, Elongation at failure 200%).*

*I believe side-by-side cyclic tests of 'just printed' and 'pre-aged' materials would need to be performed comparing the fatigue performance of jetted acrylates and thiol-enes (i.e. **Tango Black Plus and the Thiol-ene Soft**). While good **UV stability** of thiol-enes is demonstrated, this should also be a **side-by-side** comparison of an existing equivalent acrylate material. Also, **toughness tests, such as notched Izod**, should be performed to compare the performance of material jetted acrylate and thiol-enes. If these tests were to demonstrate order-of-magnitude improvement in properties, then the impact of this work would be much more apparent. Absent more robust material property characterization, I do not have the necessary data to say whether this manuscript represents a fundamental improvement over the status quo in material jetting systems.*

We recognize the lack of side-by-side tests comparing the method and material disclosed here with the state-of-the-art. The innovation of this work lies in the ability to print structures made from soft, UV stable polymers that show a reduced viscoelastic behavior. We agree that our rigid thiol-ene has comparable properties to existing rigid acrylates chemistries (e.g., Stratasys RGD 720). Yet our contribution here is the low viscosity, UV stable soft polymer that we use to create functional systems not possible with current methods. We, therefore, conducted side by side material tests comparing available soft materials as suggested by the reviewer and found order of magnitude material improvement. Specifically, when comparing materials after exposure to UV radiation according to ASTM G154. In addition to Stratasys Tango Black Plus, suggested by the reviewer, we also compared to Stratasys Agilus 30 – its successor polymer.

We were able to show that acrylate based soft polymers showed a fast deterioration of material properties when exposed to UV according to standard ASTM G154.

- For acrylates, the elongation at break was reduced to ~0.6% from an initial 130%. Our soft thiol-ene's elongation at break stayed within less than 27% of the initial value.

- The Elastic Modulus of the soft thiol-ene stayed constant at fluctuations below 0.06% compared to a 260 fold increase within 1000h of exposure for acrylate-based polymers.

We conducted side by side experiments on:

- Resilience of the material
- Solvent compatibility (weight gain)
- Dynamic mechanical analysis (DMA)
- Cyclic loading for hysteresis
- UV stability

The interfacial strength of thiol-ene prints was tested in a lap shear test.

Here are the changes made to the manuscript:

Abstract line 56ff

The advances in material properties are characterized in several standardized tests comparing material properties to state-of-the-art systems.

Line 358ff

We evaluated the materials' properties and compared it to state-of-the-art 3D jetting materials, i.e., acrylate resins (Fig. 2). Specifically, we compared soft thiol-ene to Stratasys PolyJet material Tango Black Plus. These materials match closely in terms of ultimate tensile strength (soft thiol-ene: 0.92MPa; Tango Black Plus: 0.8MPa to 1.5MPa) and elongation at break (soft thiol-ene: 200 %; Tango Black Plus: 170% to 220 %) when new. We tested the change of elastic modulus of acrylate (Tango Black Plus) and soft thiol-ene for three samples each over 1000 h in accordance with ASTM G154 [47]. ASTM G154 reproduces the weathering effects that occur when materials are exposed to sunlight and moisture (rain or dew) during real-world usage. After just 250 h, the elastic modulus of Tango Black Plus had increased by 100 fold and after 1000 h by ~260 fold. The increase was from 0.89MPa when new to 261MPa when aged. In comparison, only a ~0.06 fold change was observed for our soft thiol-ene (Fig. 2d). We also investigated the modulus of resilience of the materials according to ASTM 2632 with 3 samples per material. The acrylates were measured at 7% for Tango Black Plus and 14% for Agilus 30 (the successor of Tango Black Plus). Soft thiol-ene showed about double the modulus of resilience at 27 %. For 1000 h of exposure according to ASTM G154, our soft thiol-ene's elastic modulus stayed within 6% and elongation at break within less than 27% change compared to its value, when new (three samples were tested for each time step). In comparison, the acrylate based Tango Black plus turns brittle within less than 250 h of exposure at a reduced elongation at break of ~0.6% from initial 170% to 220% (four samples were tested for each time step). Agilus 30 showed a very similar behaviour (Fig. 2e). To quantify the viscoelastic behaviour of the materials, we recorded stress-strain cycles going from 0% to 153% displacement and back to 0% at a strain rate of ~0.53 s⁻¹. The hysteresis of the material that relates to the viscoelasticity of the material can be inferred by the area enclosed by the stress-strain cycle. We tested three samples of soft thiol-ene (area: 0.087±0.006) and Tango Black Plus (area: 0.42±0.05), each as well as two Agilus 30 (area: 0.29±0.03) samples. The acrylates' hysteresis is 3 to 4.3 times larger than the thiol-ene's hysteresis (Fig. 2f). A dynamic mechanical analysis (DMA) was conducted on soft thiol-ene (Fig. S5) and two acrylates. Thiol-ene has a much narrower region of glass transition T_g compared to the two acrylates. Changes in storage modulus were significant between -35 °C to -18 °C, whereas the two acrylate samples showed significant changes in storage modulus in a much wider range between -30 °C to 15 °C (Fig. 2g).

Line 615ff

While testing the effect of UV light, moisture, and temperature exposure to different materials, we encountered deformation, e.g., warping of some samples (Fig. S12). This led to an increased variation in the properties tested. In addition, there is no standard available to compare soft materials in different viscosity regimes. The printing technology, while widening the palette of available materials, is still limited by a relatively low viscosity of the UV-induced curing materials. The interface of prints from different material chemistries is not adhering well, yet specific tuning of the chemical composition of the materials can improve that in the future.

Figure 2 with caption

Fig. 2: Comparison of our material characteristics to the state-of-the-art. **a**, All tested materials were jetted. Jetting describes a process where ink in a nozzle is ejected in the form of droplets using a piezoelectric transducer. The droplets are deposited onto a moving substrate, e.g., a print bed. **b**, Conventional inkjet 3D printers typically use acrylate functional monomers cured using the radical polymerization (represented by molecules A, B, and C) in their inks. Chain growth polymerization results in a random and irregular distribution of monomers along the polymer backbone (blue). Since all acrylate polymers, rigid and soft, contain this common backbone chain (blue), the properties are controlled by the monomer side groups, R1 and R2, and both the structure and degree of crosslinks R3 (yellow). This combination of structural features typically results in a viscoelastic polymer. **c**, Vision controlled jetting has enabled printing of a step growth polymer (thiol-ene) using radical polymerization. The polymer chain consists of alternating thiol (D or F) and ene (E or G) monomers. The properties of the polymer are defined by the regular backbone structure and the degree of crosslinking. These properties result in an elastic polymer more suitable for robotic applications. **d**, The change in elastic modulus during 1000 h exposure according to ASTM G154 was investigated. The thiol-ene's elastic modulus was stable over time with less than ~5% change compared to the control. The acrylate, Tango Black Plus,

rigidified to its ~260 fold value within 1000 h of exposure. **e**, We also investigated the breakdown strength under exposure according to ASTM G154. Over 1000 h of exposure, we saw no relevant change in elongation at break. Both tested soft acrylates, Tango Black Plus and Agilus 30, turned brittle and broke at a minimal elongation (0.8 ± 0.2 %). **f**, The hysteresis of the material is defined by the area enclosed by a stress-strain cycle. The two acrylates tested showed 3 or 4.43 fold increased hysteresis. **g**, A dynamic mechanical analysis (DMA) was conducted on soft thiol-ene and two acrylates. Changes in storage modulus were significant between -35 °C to -18 °C, whereas the two acrylate samples showed significant changes in storage modulus in a much wider range between -30 °C to 15 °C.

Line 1432ff

The adhesion between cast soft and rigid thiol-ene was tested via lap shear ASTM D 3163-01 (Fig. S4). A shear strength of (1.08 ± 0.10) MPa was determined for five samples tested.

Line 2101ff

We tested the solvent compatibility of chemical epoxy's tensile elongation, tensile strength, and tensile modulus (Fig. S7). We compared the stability in side by side tests to state-of-the-art acrylate based materials. A rigid, acrylate based material (Stratasys RGD 720) showed warping compared to our tough epoxy (Fig. S12). In side-by-side tests with state of the art acrylates (from Stratasys) we investigated the weight gain of rigid (Tab. 5) and soft (Tab. 6) polymers under exposure to solvents. We also explored the interfacial adhesion strength by conducting a lap shear test on five samples. We adhered two bars of cast rigid thiol-ene and adhered them with a thin layer of soft thiol-ene according to ASTM D 3163-01 (Fig. S4). We observed a shear strength of (1.08 ± 0.10) MPa.

Figure S4 with caption

Fig. S4: Lap shear testing for cast soft and rigid thiol-ene. A shear strength of (1.08 ± 0.10) MPa was determined by testing five samples in accordance with ASTM D 3163-01.

Table 5 and Table 6

Table 5: Solvent compatibility: Weight gain for rigid polymers – ASTM D543

Solvent	Chem. epoxy (5-days)	Tough epoxy (7-days)	RGD 720 (7-days)
Water	0.3 wt.%	2.7 wt.%	2.5 wt.%
Acetone	0.5 wt.%	fail	fail
Isopropyl alcohol	0.0 wt.%	4.6 wt.%	3.2 wt.%
Motor Oil	0.1 wt.%	1.8 wt.%	0.6 wt.%
Ethanol	0.5 wt.%	11.5 wt.%	12.9 wt.%

Table 6: Solvent compatibility: Weight gain for soft polymers – ASTM D543

Solvent	Soft thiol-ene (7-days)	Tango Black Plus (7-days)
Water	4.3 wt.%	5.1 wt.%
Acetone	89.6 wt.%	139.7 wt.%
Isopropyl alcohol	22.9 wt.%	106.4 wt.%
Motor Oil	0.9 wt.%	5.1 wt.%
Ethanol	39.5 wt.%	fail

Remark 1G

Purely elastic soft polymers...with similar properties to those of muscles, tendons, and other complex soft tissue...". Soft tissues, such as skeletal muscle, are viscoelastic - not purely elastic.

We agree that biological soft tissue is viscoelastic rather than purely elastic. We thank the reviewers for their remark and have now corrected the statements in the text accordingly:

Line 224ff

... Without having access to ~~purely elastic (nonviscous) soft polymers~~ soft polymers with low hysteresis, it is not possible to reproduce complex functional materials and structures with similar properties ~~to those of muscles, tendons, and other complex soft tissue found in nature.~~

Figure S12 and caption

Fig. S12: Warping of acrylate based materials under exposure according to ASTM G154. Top view (a and c) and side view (b and d) images of material samples in the test setup for exposure. Our tough epoxy material (a and b) did not show warping compared to the state of the art acrylate based material (Stratasys RGD720) (c and d).

Table 5 and Table 6

Table 5: Solvent compatibility: Weight gain for rigid polymers – ASTM D543

Solvent	Chem. epoxy (5-days)	Tough epoxy (7-days)	RGD 720 (7-days)
Water	0.3 wt.%	2.7 wt.%	2.5 wt.%
Acetone	0.5 wt.%	fail	fail
Isopropyl alcohol	0.0 wt.%	4.6 wt.%	3.2 wt.%
Motor Oil	0.1 wt.%	1.8 wt.%	0.6 wt.%
Ethanol	0.5 wt.%	11.5 wt.%	12.9 wt.%

Table 6: Solvent compatibility: Weight gain for soft polymers – ASTM D543

Solvent	Soft thiol-ene (7-days)	Tango Black Plus (7-days)
Water	4.3 wt.%	5.1 wt.%
Acetone	89.6 wt.%	139.7 wt.%
Isopropyl alcohol	22.9 wt.%	106.4 wt.%
Motor Oil	0.9 wt.%	5.1 wt.%
Ethanol	39.5 wt.%	fail

Remark 1H

“Soft polymers can be smeared and squished by a roller” – how this is a disadvantage should be clarified. I.e. loss of resolution and spatially varying or mixing of material composition.

We agree with and thank the reviewers for their remark regarding the lack of detail on how smearing and squishing by a roller is a disadvantage. We have now added a description on why this hinders the printing of high-resolution functional structures.

Line 172ff

The need for mechanical planarisation ~~limits the type of materials in traditional 3D inkjet deposition~~ limits the levels of softness and the type of material chemistries that can be used ~~in a traditional 3D inkjet deposition [42]. When using mechanical planarisation, it is necessary to prevent the material~~ Soft materials cannot be reliably printed as they would be easily smeared and squished by a roller or scraper. Smearing and squishing would lead to material mixing and spatial variation of material composition and therefore a loss of printing resolution. Materials only qualify for mechanical planarisation if they can be prevented from curing on the roller or scraper. ~~Thus, it is necessary to use polymerisation processes that start upon UV irradiation and stop upon its discontinuation. For this reason, the only chemistries that can currently be used are acrylate ones. Soft polymers can be smeared and squished by a roller, and soft fast chain growth polymers are typically only acrylate chemistries relying on fast chain growth polymerisation are currently used. Fast chain growth polymerisation only occurs during UV irradiation. Unfortunately, soft acrylate chemistries are highly viscoelastic due to our poor control over the degree and structure of their crosslinking. The low printing resolution and the limitation to highly viscoelastic soft acrylates with high hysteresis hinder complex multimaterial robotic designs requiring fine soft features (such as thin tendons or fluidic channels) and rapidly deforming sections (such as soft joints) (Fig. S11).~~

Remark 11

“...Builds the polymer's structure with more precision” – this wording is unclear. A more precise discussion of step-growth benefits on degree of conversion and control over chain length and crosslinking density should be provided”.

We thank the reviewer for their remark and added a section and a major new figure (Fig. 2) describing the difference in chemistry and polymerization behavior. We specifically added a description on the benefits of step-growth polymerization. We made the following changes to the text:

Line 256ff

This slow cure mechanism builds the polymer's structure with more precision, which provides us with control over both the polymer's backbone and the degree and structure of crosslinking (Fig. 2). Acrylate resins, used in conventional 3D jetting, have only a random distribution of crosslinking polymer backbones and are therefore much less controllable in their curing status.

Figure 2 b,c with caption

b, Conventional inkjet 3D printers typically use acrylate functional monomers cured using the radical polymerization (represented by molecules A, B, and C) in their inks. Chain growth polymerization results in a random and irregular distribution of monomers along the polymer backbone (blue). Since all acrylate polymers, rigid and soft, contain this common backbone chain (blue), the properties are controlled by the monomer side groups, R₁ and R₂, and both the structure and degree of crosslinks R₃ (yellow). This combination of structural features typically results in a viscoelastic polymer. **c**, Vision controlled jetting has enabled printing of a step growth polymer (thiol-ene) using radical polymerization. The polymer chain consists of alternating thiol (D or F) and ene (E or G) monomers. The properties of the polymer are defined by the regular backbone structure and the degree of crosslinking. These properties result in an elastic polymer more suitable for robotic applications.

Response to Reviewer #2

A. Summary of the key results:

The manuscript describes a vision-based inkjet printing process, coined as “Vision-controlled jetting (VCJ)” that achieves a close-loop feedback optimization of the print layer height by capturing the topological information during printing. The contactless vision-based system removes the need for a mechanical planarizer, which expands the possible materials families that can be printed by allowing continuous curing chemistries. The authors demonstrate impressive multimaterial printing capability with several examples of multi-material actuators and architecture that are capable of integrating both rigid and soft materials.

Overall, the manuscript describes an inkjet printing platform capable of creating complex and integrated multi-material (e.g., soft and rigid polymer) constructs. However, the manuscript's scientific novelty and advancement over previous work and published public information are not clear. The manuscript also lacks the quantitative study and data needed to evaluate the key scientific claim (e.g., the quantitative data demonstrating the advancement of its closed-loop optimization in comparison to the earlier work, and mechanical properties of the multi-material structure). There are no discussions of error, potential limitations of the materials, mechanical characterization of the multi-material interface, etc as discussed below:

We like to thank the reviewer for many insightful and valuable remarks, which have enabled us to further improve the manuscript. We are glad that the reviewer agrees that we are tackling a notoriously challenging issue with material jetting systems and that we are solving it with a scalable, contactless solution. We would like to highlight that this work not only enables contactless printing, but also shows how new printing chemistries such as thiol-enes and epoxies are now possible.

In the following, we believe that we were able to address all of the reviewer's remarks.

Remark 2A

B: Originality and significance:

In regards to originality:

It is unclear what the specific conceptual novelty and originality of the proposed work are when compared to previous publications, such as published patents. For example, US10456984B2 (citation 59, 60) already describes the similar usage of “closed-loop adaptive material deposition apparatus and method uses a scanning system”. The authors did cite these patents when describing the method under “Print control software”, but did not discuss and compare the novelty and originality of their work in comparison with these advancements.

In contrast to patents, this work goes into much more detail on the methodology used, additionally provides comparative material tests and several functional multi material systems and their characterization. In our answers to the reviewer's remarks, we list the changes made to the manuscript. We added even more details on the method and in-depth characterization to the manuscript, clearly differentiating for a (non-scientific) patent publication that describes the method in general.

Remark 2B

Similarly, the idea of machine vision-assisted printing correction is not original, It was mentioned by the authors but the discussion is not well quantified. For instance, the authors claim that the previous work by Sitthi-Amorn, P. et al [ref 56] is “too slow”; but did not provide sufficient reasoning on “why & how” and the quantitative data that compares the relative difference achieved by the current work and the previous work.

Previous work from Sitthi-Amorn et al. 2015 [DOI: 10.1145/2766962] introduced a method based on optical coherence tomography scanning. The process was very slow (660 times slower) and limited to simple acrylate resins, simply inadequate for advanced functional systems. We therefore believe that the work contributed here with its new printer design, scanning system, material chemistry, and functional robotic demonstrations provides a substantial and unique advance in research. Multimaterial printed real-world systems that show complex functions are now actually feasible because of the newly allowable chemistries. We notice that this detail was not properly explained in the previous manuscript and therefore have made the following additions and changes to the revised manuscript:

Line 198ff

Previous work called MultiFab [43] relied on slow full-field optical coherence tomography (OCT) scanning of a small area of only 2 cm × 2 cm. To get information on the surface structure of the full print surface, it had to be rasterised and each area had to be scanned. Next, all scanned areas had to be stitched together in post-processing. Only then could the corrections for the next layer be computed. This post-processing leads to a limited throughput of only 0.05mLmin⁻¹ and print times per layer on the order of minutes, even for smaller prints. MultiFab therefore cannot keep up with inkjet deposition speeds and does not scale to produce large functional parts such as the ones presented in this work. Furthermore, previous work [43] only supported simple acrylate resins, which are inadequate for advanced functional systems.

Line 256ff

Our method utilizes a high-speed 3D vision system to capture a depth map of the currently printed surface, and it compensates for deviations from an ideal planar surface by locally adapting the amount of material to be jetted in the next layer. The method's vision system uses four cameras and two laser sources for laser profilometry while printing. The feedback-loop including the surface scan of the whole print area are performed at printing speed and therefore do not slow down the print process. This leads to a throughput of up to 33mLmin⁻¹, which is 660 times faster than the previous work [43].

Line 984ff

The integrated systems and robots presented in this work were all 3D printed using a multimaterial additive manufacturing platform that utilised a vision-controlled jetting technology. ~~Previous work [43] relied on slow optical coherence tomography (OCT) scanning, which cannot keep up with inkjet deposition speeds and does not scale to produce large functional parts such as the ones presented in this work. Furthermore, previous work [43] only supported simple acrylate resins, which are inadequate for advanced functional systems.~~

Remark 2C

It also appears that most of the novelty presented here has already been commercialized and published on some of the authors' company websites (inkbit3d.com), which was not explicitly referenced and highlighted. In fact, some of the videos on the website (<https://inkbit3d.com/technology/>) are identical to the videos that have been submitted for consideration in this manuscript. It is unclear what is the original and novel conceptual and scientific advancement that is being discussed in this manuscript in comparison to public information, especially when many of the claims (e.g., multimaterial adhesion strength, materials compatibilities) are not substantiated with experimental characterizations, but are limited to visual proof of concept.

Some material data was previously available online, and a rendered overview of the printing process is available on the website. However, the bulk of test data, specifically the side-by-

side comparisons added in this revision, the multimaterial systems, and their characterization are novel. We now provide a detailed section with information on the materials:

We were able to demonstrate that acrylate based soft polymers showed a fast deterioration of material properties when exposed to UV according to standard G154.

- Elongation at Break was reduced for acrylates to $\sim 0.6\%$ from initial 130%. Our soft thiol-ene's elongation at break stayed within less than 27% of the initial value.
- The Elastic Modulus of the soft thiol-ene stayed constant at fluctuations below 0.06% compared to a 260-fold increase within 1000h of exposure for acrylate based polymers.

We conducted side by side experiments on:

- Resilience of the material
- Solvent compatibility (Weight gain)
- Dynamic mechanical analysis (DMA)
- Cyclic loading for Hysteresis
- UV stability

The interfacial strength of thiol-ene prints was tested in a lap shear test.

Here are the changes made to the manuscript:

Abstract line 56ff

The advances in material properties are characterized in several standardized tests comparing material properties to state-of-the-art systems.

Line 358ff

We evaluated the materials' properties and compared it to state-of-the-art 3D jetting materials, i.e., acrylate resins (Fig. 2). Specifically, we compared soft thiol-ene to Stratasys PolyJet material Tango Black Plus. These materials match closely in terms of ultimate tensile strength (soft thiol-ene: 0.92MPa; Tango Black Plus: 0.8MPa to 1.5MPa) and elongation at break (soft thiol-ene: 200 %; Tango Black Plus: 170% to 220 %) when new. We tested the change of elastic modulus of acrylate (Tango Black Plus) and soft thiol-ene for three samples each over 1000 h in accordance with ASTM G154 [47]. ASTM G154 reproduces the weathering effects that occur when materials are exposed to sunlight and moisture (rain or dew) during real-world usage. After just 250 h, the elastic modulus of Tango Black Plus had increased by 100 fold and after 1000 h by ~ 260 fold. The increase was from 0.89MPa when new to 261MPa when aged. In comparison, only a ~ 0.06 fold change was observed for our soft thiol-ene (Fig. 2d). We also investigated the modulus of resilience of the materials according to ASTM 2632 with 3 samples per material. The acrylates were measured at 7% for Tango Black Plus and 14% for Agilus 30 (the successor of Tango Black Plus). Soft thiol-ene showed about double the modulus of resilience at 27 %. For 1000 h of exposure according to ASTM G154, our soft thiol-ene's elastic modulus stayed within 6% and elongation at break within less than 27% change compared to its value, when new (three samples were tested for each time step). In comparison, the acrylate based Tango Black plus turns brittle within less than 250 h of exposure at a reduced elongation at break of $\sim 0.6\%$ from initial 170% to 220% (four samples were tested for each time step). Agilus 30 showed a very similar behaviour (Fig. 2e). To quantify the viscoelastic behaviour of the materials, we recorded stress-strain cycles going from 0% to 153% displacement and back to 0% at a strain rate of $\sim 0.53 \text{ s}^{-1}$. The hysteresis of the material that relates to the viscoelasticity of the material can be inferred by the area enclosed by the stress-strain cycle. We tested three samples of soft thiol-ene (area: 0.087 ± 0.006) and Tango Black Plus (area: 0.42 ± 0.05), each as well as two Agilus 30

(area: 0.29 ± 0.03) samples. The acrylates' hysteresis is 3 to 4.3 times larger than the thiol-ene's hysteresis (Fig. 2f). A dynamic mechanical analysis (DMA) was conducted on soft thiol-ene (Fig. S5) and two acrylates. Thiol-ene has a much narrower region of glass transition T_g compared to the two acrylates. Changes in storage modulus were significant between -35 °C to -18 °C, whereas the two acrylate samples showed significant changes in storage modulus in a much wider range between -30 °C to 15 °C (Fig. 2g).

Line 615ff

While testing the effect of UV light, moisture, and temperature exposure to different materials, we encountered deformation, e.g., warping of some samples (Fig. S12). This led to an increased variation in the properties tested. In addition, there is no standard available to compare soft materials in different viscosity regimes. The printing technology, while widening the palette of available materials, is still limited by a relatively low viscosity of the UV-induced curing materials. The interface of prints from different material chemistries is not adhering well, yet specific tuning of the chemical composition of the materials can improve that in the future.

Figure 2 with caption

Fig. 2: Comparison of our material characteristics to the state-of-the-art.] a, All tested materials were jetted. Jetting describes a process where ink in a nozzle is ejected in the form of droplets using a piezoelectric transducer. The droplets are deposited onto a moving substrate, e.g., a print bed. b, Conventional inkjet 3D printers typically use acrylate functional monomers cured using the radical polymerization (represented by molecules A, B, and C) in their inks. Chain growth polymerization results in a random and irregular distribution of monomers along the polymer backbone (blue). Since all acrylate polymers, rigid and soft, contain this common backbone chain (blue), the properties are controlled by the monomer side groups, R1 and R2, and both the structure and degree of crosslinks R3 (yellow). This combination of structural features typically results in a viscoelastic polymer. c, Vision controlled jetting has

enabled printing of a step growth polymer (thiol-ene) using radical polymerization. The polymer chain consists of alternating thiol (D or F) and ene (E or G) monomers. The properties of the polymer are defined by the regular backbone structure and the degree of crosslinking. These properties result in an elastic polymer more suitable for robotic applications. **d**, The change in elastic modulus during 1000 h exposure according to ASTM G154 was investigated. The thiol-ene's elastic modulus was stable over time with less than ~5% change compared to the control. The acrylate, Tango Black Plus, rigidified to its ~260 fold value within 1000 h of exposure. **e**, We also investigated the breakdown strength under exposure according to ASTM G154. Over 1000 h of exposure, we saw no relevant change in elongation at break. Both tested soft acrylates, Tango Black Plus and Agilus 30, turned brittle and broke at a minimal elongation ($(0.8 \pm 0.2) \%$). **f**, The hysteresis of the material is defined by the area enclosed by a stress-strain cycle. The two acrylates tested showed 3 or 4.43 fold increased hysteresis. **g**, A dynamic mechanical analysis (DMA) was conducted on soft thiol-ene and two acrylates. Changes in storage modulus were significant between $-35 \text{ }^\circ\text{C}$ to $-18 \text{ }^\circ\text{C}$, whereas the two acrylate samples showed significant changes in storage modulus in a much wider range between $-30 \text{ }^\circ\text{C}$ to $15 \text{ }^\circ\text{C}$.

Line 1432ff

The adhesion between cast soft and rigid thiol-ene was tested via lap shear ASTM D 3163-01 (Fig. S4). A shear strength of $(1.08 \pm 0.10)\text{MPa}$ was determined for five samples tested.

Line 2101ff

We tested the solvent compatibility of chemical epoxy's tensile elongation, tensile strength, and tensile modulus (Fig. S7). We compared the stability in side by side tests to state-of-the-art acrylate based materials. A rigid, acrylate based material (Stratasys RGD 720) showed warping compared to our tough epoxy (Fig. S12). In side-by-side tests with state of the art acrylates (from Stratasys) we investigated the weight gain of rigid (Tab. 5) and soft (Tab. 6) polymers under exposure to solvents. We also explored the interfacial adhesion strength by conducting a lap shear test on five samples. We adhered two bars of cast rigid thiol-ene and adhered them with a thin layer of soft thiol-ene according to ASTM D 3163-01 (Fig. S4). We observed a shear strength of $(1.08 \pm 0.10)\text{MPa}$.

Figure S4 with caption

Fig. S4: Lap shear testing for cast soft and rigid thiol-ene. A shear strength of $(1.08 \pm 0.10)\text{MPa}$ was determined by testing five samples in accordance with ASTM D 3163-01.

Table 5 and Table 6

Table 5: Solvent compatibility: Weight gain for rigid polymers – ASTM D543

Solvent	Chem. epoxy (5-days)	Tough epoxy (7-days)	RGD 720 (7-days)
Water	0.3 wt.%	2.7 wt.%	2.5 wt.%
Acetone	0.5 wt.%	fail	fail
Isopropyl alcohol	0.0 wt.%	4.6 wt.%	3.2 wt.%
Motor Oil	0.1 wt.%	1.8 wt.%	0.6 wt.%
Ethanol	0.5 wt.%	11.5 wt.%	12.9 wt.%

Table 6: Solvent compatibility: Weight gain for soft polymers – ASTM D543

Solvent	Soft thiol-ene (7-days)	Tango Black Plus (7-days)
Water	4.3 wt.%	5.1 wt.%
Acetone	89.6 wt.%	139.7 wt.%
Isopropyl alcohol	22.9 wt.%	106.4 wt.%
Motor Oil	0.9 wt.%	5.1 wt.%
Ethanol	39.5 wt.%	fail

Remark 2D*Breadth of claim and significance:*

While the proposed approach appears to be a significant advancement of the current multijet printing approach, some of the claims are not specific and stretch too far beyond. For instance, the claim that the first step of the “organism that mimics nature”, as the demonstrated example remains limited to a few selected materials compatible with the inkjet printing process.

We agree with the reviewer on some of the claims being not specific enough and may stretch too far beyond. We have now fixed this in the revision. While we adapted the abstract and the introduction, we did not find overreaching claims in the conclusion. In case we missed such a claim, we would be thankful if they are pointed out. Here is how we adapted the abstract and the introduction:

Abstract

Recreating the complex structure and function of natural organisms in a synthetic form is a long-standing goal for humanity [1]. Natural organisms have a high spatial resolution and complex material arrangements that range from elastic to rigid [2]. Their actuation mechanisms and sensing receptors are seamlessly integrated into their structures. Traditional manufacturing processes do not allow for the fabrication of such complex systems. Despite remarkable progress [3, 4], it remains an open challenge to fabricate functional systems automatically and quickly with a wide range of elastic properties, resolutions comparable to natural systems, and integrated actuation and sensing. We propose an inkjet deposition process called Vision-Controlled Jetting (VCJ), and we show its capability through examples of composite systems and robots. A scanning system captures the three-dimensional print geometry and enables a digital feedback loop, which eliminates the need for mechanical planarizers. This contactless process allows us to use continuously curing chemistries and, therefore, print a wide range of material families and elastic moduli into complex high-resolution composite structures. The advances in material properties are characterized in several standardized tests comparing material properties to state-of-the-art systems. We have

directly fabricated a wide range of composite systems and robots with integrated actuation and sensing: a tendon-driven hand adapted from magnetic resonance imaging (MRI) data [5], a pneumatically actuated walking manipulator, a pump that mimics a heart, and metamaterial structures. Our approach provides an automated, scalable, high-throughput process to manufacture high-resolution, functional multimaterial systems with integrated sensing and actuation.

Introduction, line 216ff

Without having access to ~~purely elastic (nonviscous) soft polymers with low hysteresis,~~ it is not possible to reproduce complex functional materials and structures with similar properties ~~to those of muscles, tendons, and other complex soft tissue found in nature.~~

Remark 2E

C: Data & methodology: validity of approach, quality of data, quality of presentation

The authors did not adequately relate or perform the data and characterization that are critical to understanding the core proposed novelty, especially when comparing the advancement of this work with the earlier work.

We agree that we did not sufficiently discuss the difference to earlier works and adapted the manuscript to discuss these in more detail. A detailed reply to this remark with all changes to the manuscript listed is provided in the answer for **Remark 2B**.

Remark 2F

For instance, the accuracy of the scanning, and the accuracy of the print compensation are not discussed in the main text.

We already provided the resolution of the scanner and accuracy of the prints in the main text.

Line 307f

The 3D vision system provides highly accurate depth maps of the surface at a resolution of $64\ \mu\text{m} \times 32\ \mu\text{m} \times 8\ \mu\text{m}$.

Table 7

Table 7: Accuracy for prints with tough epoxy. – ISO/ASTM 52902

	Linear accuracy – Part length		Circular accuracy – Diameter				
	0 mm to 25 mm	25 mm to 55 mm	7 mm	8 mm	15 mm	23.5 mm	25 mm
X&Y	± 0.1	± 0.1	-0.03	0.10	0.01	0.13	0.11
Z	± 0.2	± 0.3	-0.1	0.14	-0.1	0.07	0.02

Remark 2G

The Tables in supporting information could have been better described and related to the main text.

We agree with the reviewer's remark and now reference supporting information better in the main text. We now relate in the main text to the tables and figures of the supporting text.

Remark 2H

Further, a core part of the manuscript's novelty is its multi-material capability: but the critical information such as the interfacial strength, and fatigue strength of the multi-material interface are not discussed, which limits the understanding of the application of the described approach.

We conducted lap shear tests on the multi material constructs. Samples are made of a thin layer of soft thiol-ene sandwiched between two rigid thiol-ene elements. We conducted lap shear testing. This test allows us to determine the adhesion of the two materials to each other. The results are shown in supplementary figure S4.

Line 2117ff

We also explored the interfacial adhesion strength by conducting a lap shear test on five samples. We adhered two bars of cast rigid thiol-ene and adhered them with a thin layer of soft thiol-ene according to ASTM D 3163-01 (Fig. S4). We observed a shear strength of (1.08 ± 0.10) MPa.

Figure S4 with caption

Fig. S4: Lap shear testing for cast soft and rigid thiol-ene. A shear strength of (1.08 ± 0.10) MPa was determined by testing five samples in accordance with ASTM D 3163-01.

Remark 2I

D: Appropriate use of statistics and treatment of uncertainties

There is insufficient statistical information provided in some of the data (e.g., Tables in supporting information) to evaluate the uncertainty and robustness of the presented data. There is no discussion on the sample size, the standard deviation of the data, etc.

We agree with the reviewer's remark and now add sample sizes and the standard deviation for all our test data on material properties where more than one test was conducted.

Remark 2J

E: Conclusions: robustness, validity, reliability

Additional experimental data, analysis, and discussion can help quantitatively evaluate the robustness, validity, and reliability of the robustness of several key results, such as the strength of the multimaterial structure; particularly when comparing the attributes with the earlier approach.

We agree that we did not sufficiently discuss the difference to earlier works and an evaluation of the multimaterial structure's strength. We adapted the manuscript to discuss these in more detail. A detailed reply to this remark with all changes to the manuscript listed is provided in the answer for **Remark 2B** and **Remark 2H**.

Remark 2K

F: Suggested improvements: experiments, data for possible revision

Several suggestions, as described above, are provided below:

- The novelty of the proposed work can be quantitatively compared with the earlier work and published information. E.g., when discussing another approach as "too slow", what is the relative speed improvement that has been achieved, and how?

We agree that we did not sufficiently discuss the difference to earlier works and adapted the manuscript to discuss these in more detail. A detailed reply to this remark with all changes to the manuscript listed is provided in the answer for **Remark 2B**.

Remark 2L

- The novelty of the work can be more clear if it is focused on what are the specific contributions of the work – e.g., if the advancement is on the scanning strategy – what are the scientific advancements here that achieve what was not possible?

We agree with the reviewer's remark on the lack of clarity and focus of the description of our contribution and adapted the manuscript accordingly.

Line 261ff

Acrylate resins, used in conventional 3D jetting, have only a random distribution of crosslinking polymer backbones and are therefore much less controllable in their curing status.

Line 282ff

The printer's build volume is 500mm × 245mm × 200 mm. The print heads and the print speed are therefore on-par with state-of-the-art inkjet printers for this scale and resolution but our print system including the scanner allows for the first time a wider range of chemistries and therefore material properties compared to other printers.

Line 292ff

In this method, droplets of ~~multiple~~ different types of resins ~~as build material~~ (build material) are jetted concurrently with droplets of wax ~~as support material~~ (support material). The resin then polymerises using UV-radiation and the wax solidifies when cooled down. ~~The process~~ We can use both fast and slow chain growth polymer chemistries with our contactless approach. Similar to conventional inkjet printers, our process also utilizes piezoelectric print heads (4 print heads per material, 1024 nozzles per print head), ~~allowing~~ but we additionally introduce a fast surface profilometer. The closed-loop controlled system allows for single-pass multimaterial printing with currently up to three build materials and one support material (Fig. S1). ~~This approach leads to much faster layer by layer printing than other printing methods, which need to first build each layer line by line.~~

Remark 2M

- There is an inadequate discussion on what the limitation of the material is with the proposed approach, and how that impacts its potential application.

We agree with the reviewer's remark and added a more in depth discussion on the limitations of our approach.

Line 615ff

While testing the effect of UV light, moisture, and temperature exposure to different materials, we encountered deformation, e.g., warping of some samples (Fig. S12). This led to an increased variation in the properties tested. In addition, there is no standard available to compare soft materials in different viscosity regimes. The printing technology, while widening the palette of available materials, is still limited by a relatively low viscosity of the UV-induced curing materials. The interface of prints from different material chemistries is not adhering well, yet specific tuning of the chemical composition of the materials can improve that in the future. Additionally, the high resolution of the printer allows for the printing of features for mechanical interlocking of these interfaces. The availability of four print heads still limits the complexity of materials that can be printed in one process. The print process is inherently based on the ample application of support material. While it is easily liquefied and removable (Fig. S2), every created cavity has to have a connection to the outside of the part for

drainage. Additionally, the removal of liquid support material from small or porous sponge-like structures is still difficult due to the high surface tension and despite the use of surfactants to lower the surface tension.

Remark 2N

- The claim of the broadness of impact can be more specific and relevant to the demonstrated data.

We agree with the reviewer's remark and replied in detail to this in our answer to **Remark 2D**.

Remark 2O

- Critical data describing the fundamental attribute can be made the focus of the work instead of extensive figures that focuses primarily on different applications. While the applications are impressive - it does not adequately provide the quantitative information needed to understand the strength, uniqueness, and limitation of the approach.

We agree with the reviewer's remark on the lack of quantitative information on the materials presented here. We provide a detailed reply in our answer to **Remark 2C**.

Remark 2P

G: References: appropriate credit to previous work?:

Yes, the majority of the reference is appropriate

However, some citations are not discussed in the introduction, and some of the submitted materials are publicly available material (published on the author's company website) but are not referenced in the manuscript. It is unclear which content is made original to the manuscript, and which content is a reproduction of the authors' published earlier work.

Previously on Inkbit's website, we disclosed some of the average property values for soft thiol-ene, tough epoxy, and chemical epoxy. This older data, stemming from a preliminary material analysis, was made available to investors and interested parties upon request so these parties could better evaluate the Inkbit company (commercializing the VCJ technology). The online availability of this information was also previously disclosed to the editor who did not raise any issue with this divulgation.

In this manuscript, we provide far more extensive experimental data and characterizations as also shown in our detailed reply to **Remark 2C**.

Remark 2Q

H: Clarity and context: lucidity of abstract/summary, appropriateness of abstract, introduction and conclusions:

The abstract, introduction, and conclusion can be revised to be more specific to the advancement that is made. The claim that this is the "first step to quickly and reproducibly create organisms that mimic nature" seems too far-fetched, especially when the material limitation here is not adequately described.

We agree with the reviewer and adapted the manuscript. A detailed reply to this remark can be found in our answer to **Remark 2D**.

Remark 2R

Further, it is not clear whether the highlighted attribute, such as the jetting speed and the number of nozzles is the unique core novelty of the work. If the jetting speed and the number

of nozzles have already been achieved by the previous inkjet printing approach, highlighting these as a key attribute can cause confusion to the general reader who may not be familiar with inkjet printing on what are the key advancement of the work.

We agree with the reviewer that we did not attribute the novelty of the work well. The print head, its resolution, and the print speed are not a contribution of this work. We adapted the manuscript to clarify this better:

Line 282ff

The printer's build volume is 500mm × 245mm × 200 mm. The print heads and the print speed are therefore on-par with state-of-the-art inkjet printers for this scale and resolution but our print system including the scanner allows for the first time a wider range of chemistries and therefore material properties compared to other printers.

Remark 2S

The authors have included many beautifully crafted figures of the 3D models and impressive demonstrations with the 3D printed prototypes. However, many of the presentations of the figures, especially those that involve quantitative data can be improved. Many of the fonts are too small and difficult to read, and some of the labels overlap with the grid line.

We understand that the text in some figures is small. Nature's guidelines regarding final artwork limit the minimum text size to 5pt and the maximum to 7pt. Most text in our figures is 7pt with only minor parts being 6.5pt or smaller. We carefully checked and are not aware of any text in the figures that is smaller than the allowed 5pt. However, if there are specific portions of figures that you would like to see enlarged, we are happy to make that change.

Remark 2T

Minor comments: There are a few typos: for example pg 27 – the first quotation mark for "forearm" is inverted.

We corrected the inverted quotation mark on page 27 and once again proofread the entire manuscript several times. We found a few further typos that we have now corrected.

Reviewer Reports on the First Revision:

Referees' comments:

Referee #1 (Remarks to the Author):

The authors have provided substantial new data to address my previous concerns that centered on whether the thiol-ene (in particular the soft thiol-ene) offered significant advantages over acrylate-based chemistries.

The new data demonstrates significant improvements in resilience, UV/moisture/temperature exposure, and solvent exposure. In light of the new data, I recommend the manuscript for publication, and offer the following minor comments:

This statement is vague and should specify why acrylate resins are inadequate: "previous work [43] only supported simple acrylate resins, which are inadequate"

Page 9, Line 370 & Page 17: Line 773: I recommend changing 'Exposure' to 'UV exposure' for increased clarity.

Page 9, Line 382: Please clarify whether resilience measurements were performed before or after UV/moisture exposure.

Please provide brief details on how the 1000h post-exposure was performed in methods.

Referee #2 (Remarks to the Author):

In the revised manuscript, the authors have addressed the majority of the previous comments. Based on the revised manuscript, several remaining comments and suggestions are proposed. Primarily, the core novelty of the work can perhaps be made clearer in the manuscript, which can potentially better highlight the key results, originality, and significance of the work.

1. The comparison to "nature" seems confusing and may be misleading. While the manuscript is an impressive advancement in material jetting additive manufacturing system (via the integration of scanning system and ink), it is far from "recreating the complex structure and function of natural organisms" as implied. While the authors have demonstrated the ability to create more complex structures in comparison to previous material jetting, what has been demonstrated remains primarily structural materials from a relatively few classes of materials; in stark contrast to the highly complex & intrinsically multifunctional & multi-materials demonstrated in nature.

2. Also importantly, as noted in the previous reviewer's comment, such proof-of-concept demonstrations have already been demonstrated with previous material jetting systems, though with less sophistication. The fundamental contribution is not on the first multi-material integration with material jetting, but on advancement in the key limitation that improved what has been accomplished earlier. While the authors have now made reference to these systems in the revised text, the attributions and discussion of key advancements in comparison to these works should be made clearer in the abstract and introduction. For example, the current abstract remains confusing to general readers – as it seems to imply that this is the first time such multi-material has been achieved with reference to a "natural organism". It is suggested that perhaps the authors can consider removing references to the "natural organism" from the abstract and introduction. Instead, focus directly on the specific advancement made in the materials jetting system which the authors have made in the later paragraphs.

3. Similarly, the claim that these have “integrated sensing and actuation” are potentially misleading and confusing and should be removed. The author’s printed parts only include part of the passive components as part of the sensing and actuation in the corresponding proof of concept examples. However, what has been demonstrated so far is incapable of printing what is typically referred to as an “embedded sensor or actuator”. The demonstrated system in the manuscript relies on external drivers and electronics (which are not embedded). While what is demonstrated is a step forward as an integrated passive mechanical component and indeed is a step closer to creating a highly complex structure, it is far from achieving “integrated sensing and actuation” as what seems to be implied with what the natural organism has achieved.

4. The comparison with the direct ink writing (DIW) approach may not be an apple-to-apple comparison. DIW and Material Jetting are distinct classes of additive manufacturing technologies that have their unique attributes. It is well-known that material jetting can have higher throughput and achieve better multi-material integration but is usually limited with inks with similar chemical basis. Based on the revised manuscript, while a significant advances over the previous material jetting system, it still lacks the versatility of the range of materials that a DIW has shown due to the fundamentally limited compatibility of the ink for the jetting process (e.g., orders of magnitude narrower viscosity range). While contrasting the advancement made in this work with material jetting makes sense, the current comparison with DIW, (which seems to suggest that the proposed system will supplant a DIW system) should be clarified; or further justification and demonstration of how the system can overcome these fundamental limitations in comparison to DIW should be demonstrated.

5.(Minor comment)

Some of the descriptions are vague and can be quantified further. For example, the authors stated that:

"A rapid and accurate multimaterial additive manufacturing method is required to repeatably produce hybrid soft rigid systems with a fine resolution at scale."

It will be much more meaningful if the authors can provide a quantitative description of what they mean by rapid, accurate, hybrid, and fine resolution.

6 (Minor comment)

The revised manuscripts have improved their presentations and have clearer descriptions and clearer illustrations. However, a few typos and inconsistencies remain. For example, in the legend of Figure 2, some are labeled as thiol-ene, and some are labeled as thiolene. Some of the character 'l' is bolded. While the font size and format meet the minimum requirement of the journal, the graph of Figure 2 can be difficult to read due to the choice of font, the color of the marker, and alignment. While it is readable, it seems to have room for improvement when comparing the quality of the figure with other papers published in this journal.

Author Rebuttals to First Revision:

Title: Vision-Controlled Jetting for Composite Systems and Robots

Authors: Thomas J. K. Buchner, Simon Rogler, Stefan Weirich, Yannick Armati, Barnabas Gavin Cangan, Javier Ramos, Davide Marini, Aaron Weber, Desai Chen, Scott Twiddy, Joshua Jacob, Greg Ellson, Walter Zengerle, Dmitriy Katalichenko, Chetan Keny, Wojciech Matusik^c and Robert K. Katzschmann^c

Response to Referee #1

The authors have provided substantial new data to address my previous concerns that centered on whether the thiol-ene (in particular the soft thiol-ene) offered significant advantages over acrylate-based chemistries.

The new data demonstrates significant improvements in resilience, UV/moisture/temperature exposure, and solvent exposure. In light of the new data, I recommend the manuscript for publication, and offer the following minor comments:

We like to thank the referee for their remarks that enabled us to improve the manuscript significantly. We are glad that we were able to address the referee's previous concerns by providing substantial new data. In the following, we would like to address the referee's remaining minor comments.

Remark 1A

This statement is vague and should specify why acrylate resins are inadequate: "previous work [43] only supported simple acrylate resins, which are inadequate"

We agree with the referee and add statements to clarify the shortcomings of acrylate resins regarding applications.

Line 88f: [...] Especially soft acrylates are not stable to the environment and their deformation behaviour is dominated by hysteresis. Without access to soft polymers with low hysteresis, it is not possible to reproduce complex functional materials and structures with desirable properties. Printing hybrid soft-rigid systems necessitates functional polymers that can crosslink in a controlled manner to minimize viscoelasticity while achieving a wide range of stiffnesses. [...]

Remark 1B

Page 9, Line 370 & Page 17: Line 773: I recommend changing 'Exposure' to 'UV exposure' for increased clarity.

We agree with the referee's remarks about increasing the clarity of the statements regarding the ageing of the material. Yet, instead of changing exposure to UV exposure (that would omit the information on the exposure to temperature and humidity) we now use the term "outdoor weathering" and describe the procedure in more detail in the methods section.

Line 160ff: We tested the change of elastic modulus of acrylate (Tango Black Plus) and soft thiol-ene when exposed to outdoor weathering including UV exposure, temperature changes, and humidity (ASTM G154).

Fig. 2 caption: d, The change in elastic modulus during outdoor weathering for Tango Black Plus and soft thiol-ene.

Line 727ff: 2.4 Outdoor weathering using ASTM G154 Cycle 1

ASTM G154 mimics outdoor weathering in addition to UV exposure. It reproduces the weathering effects that occur when materials are exposed to sunlight and moisture (rain or dew) during real-world usage. Rather than just exposure humidity this test causes water droplets to form on the parts' surface, modelling dew formation.

The testing standard ASTM G154 cycle 1 exposes all samples to 0.89 W/(m² nm) UV irradiation at a wavelength of about 340 nm from a UVA-340 lamp. The exposure cycle consists of 8 h UV at (60 ±3) °C Black Panel Temperature followed by 4 h Condensation at (50±3) °C Black Panel Temperature. The test samples were removed and tested after 250 h, 500 h, 750 h and 1000 h.

Remark 1C

Page 9, Line 382: Please clarify whether resilience measurements were performed before or after UV/moisture exposure.

We thank the referee for this remark and added a section in Methods providing details on the measurements.

Line 711ff: **2.1 Modulus of resilience using ASTM 2632**

We investigated the modulus of resilience of the materials directly from the printer according to ASTM 2632 [61] with 3 samples per material. ASTM 2632 specifies the test parameters for impact resilience of solid rubber from the measurement of the vertical rebound of a dropped mass from 16 inches.

Remark 1D

Please provide brief details on how the 1000h post-exposure was performed in methods.

We thank the referee for this remark and added a section in the methods that gives details on all tests performed for the characterization.

Line 708ff: **2 Testing standards and material characterization**

We used standardized testing to evaluate the printable materials compared to the state-of-the-art materials. In the following, we describe the standards used in this work. [...]

Line 727ff: **2.4 Outdoor weathering using ASTM G154 Cycle 1**

ASTM G154 [63] mimics outdoor weathering in addition to UV exposure. The test reproduces the weathering effects that occur when materials are exposed to sunlight and moisture (rain or dew) during real-world usage. Rather than just exposure humidity, this test causes water droplets to form on the parts' surface, modelling dew formation.

The testing standard ASTM G154, Cycle 1 exposes all samples to 0.89 W/(m² nm) UV irradiation at a wavelength of about 340 nm from a UVA-340 lamp. The exposure cycle consists of 8 h UV at (60 ±3) °C Black Panel Temperature followed by 4 h Condensation at (50±3) °C Black Panel Temperature. The test samples were removed and tested after 250 h, 500 h, 750 h and 1000 h.

Response to Referee #2

In the revised manuscript, the authors have addressed the majority of the previous comments. Based on the revised manuscript, several remaining comments and suggestions are proposed. Primarily, the core novelty of the work can perhaps be made clearer in the manuscript, which can potentially better highlight the key results, originality, and significance of the work.

We like to thank the referee for their remarks that enabled us to improve the manuscript substantially. We are glad that we were able to address the referee's previous concerns by providing substantial new data. In the following, we would like to address the referee's remaining minor comments, including how we further clarified the core novelty of the work and better highlighted the key results, originality, and significance of the work.

Remark 2A

1. The comparison to “nature” seems confusing and may be misleading. While the manuscript is an impressive advancement in material jetting additive manufacturing system (via the integration of scanning system and ink), it is far from “recreating the complex structure and function of natural organisms” as implied. While the authors have demonstrated the ability to create more complex structures in comparison to previous material jetting, what has been demonstrated remains primarily structural materials from a relatively few classes of materials; in stark contrast to the highly complex & intrinsically multifunctional & multi-materials demonstrated in nature.

We agree with the referee and removed or toned down our comparisons to nature. While we are inspired by nature and aim to be able to fabricate structures at a similar level of sophistication in the future, we are not able to do so yet. We re-wrote parts in question in the summary paragraph, the introduction, and the discussion.

Line 16ff (Abstract):

Recreating complex structures and functions of natural organisms in a synthetic form is a long-standing goal for humanity [1]. The aim is to create actuated systems with high spatial resolutions and complex material arrangements that range from elastic to rigid. Traditional manufacturing processes struggle to fabricate such complex systems [2]. It remains an open challenge to fabricate functional systems automatically and quickly with a wide range of elastic properties, resolutions, and integrated actuation and sensing channels [2, 3]. [...]

Line 43ff (First intro paragraph):

[...] Recently developed hybrid soft-rigid systems can already outperform rigid systems in certain unstructured environments [16,17] by adapting to unknown situations [18] and interacting with living beings in a safe manner [19]. In addition, we must include channels and cavities to carry, for example, signals, power, or materials. These features are important but difficult to replicate.

Remark 2B

2. Also importantly, as noted in the previous reviewer's comment, such proof-of-concept demonstrations have already been demonstrated with previous material jetting systems, though with less sophistication. The fundamental contribution is not on the first multi-material integration with material jetting, but on advancement in the key limitation that improved what has been accomplished earlier. While the authors have now made reference to these systems in the revised text, the attributions and discussion of key advancements in comparison to these works should be made clearer in the abstract and introduction.

We have further clarified the key advancements in comparison to the previous works. Here are some examples:

Line 62ff:

Traditional 3D inkjet printing uses thousands of individually addressable nozzles to deposit **low-viscosity** resins that are mechanically planarized and UV cured [30]. For a comparable resolution, inkjet deposition leads to orders-of-magnitude faster layer-by-layer printing than other line-by-line printing methods (*e.g.*, DIW or fused filament fabrication). Traditional 3D inkjet prints multi-material bellows that can be assembled to suction grippers [31], intersperses

inks to create discrete changes in material stiffness [32], turns soft and rigid acrylates into thin layers of shape memory polymers [33], and jets also non-curing inks to create hydraulic systems [34, 35].

Line 70ff: Inkjet droplet deposition varies in ink volume due to a variable flow rate and nozzle cross-talk. Therefore, each printed layer requires mechanical planarization, which limits the levels of softness and the type of material chemistries that can be used [36]. Soft or slow-curing materials would be easily smeared and squished by a roller or scraper, leading to material mixing and spatial variation of material composition. Materials only qualify for mechanical planarization if they can be prevented from curing on the roller or scraper. For this reason, 3D inkjet deposition currently only uses acrylate chemistries that rely on fast chain-growth polymerisation that only occurs during UV irradiation. [...]

Line 90ff: Without access to soft polymers with low hysteresis, it is not possible to reproduce complex functional materials and structures with desirable properties. Printing hybrid soft-rigid systems necessitates functional polymers that can crosslink in a controlled manner to minimize viscoelasticity while achieving a wide range of stiffnesses. Complex functional systems also require cavities and channels of fine resolution across large build volumes despite a high print throughput. These desirable material chemistries and structural features can be realized if we employ a non-contact planarization strategy and allow for easily removable support materials (such as wax) (Extended Data Fig. 2).

Line 98ff: **Vision Controlled Jetting**
Here, we present a method for inkjet-based multi-material deposition that expands the range of printable materials and the degree of material hardness by means of contactless, continuous print adjustments (Supplementary Video 1). We call this manufacturing method Vision Controlled Jetting (VCJ) (Fig. 1 and Supplementary Video 2). Our method utilizes a high-speed 3D vision system to capture a depth map of the currently printed surface, and it compensates for deviations from an ideal planar surface by locally adapting the amount of resin to be jetted in the next layer. The method's vision system uses four cameras and two laser sources for laser profilometry while printing. The feedback loop including the surface scan of the whole print area is performed without slowing down the print process. Our method is 660 times faster than previous work [43] by achieving a throughput of up to 33 mL/min.

Remark 2C

For example, the current abstract remains confusing to general readers – as it seems to imply that this is the first time such multi-material has been achieved with reference to a “natural organism”. It is suggested that perhaps the authors can consider removing references to the “natural organism” from the abstract and introduction. Instead, focus directly on the specific advancement made in the materials jetting system which the authors have made in the later paragraphs.

We thank the referee for the feedback and minimized references to “natural organisms”. For details, please see our answer to Remark 2A for more details on how we removed some reference on natural organisms where it was possibly misleading and see our answer to Remark 2B on how we have now focused on our work's specific advancements.

Remark 2D

3. Similarly, the claim that these have “integrated sensing and actuation” are potentially misleading and confusing and should be removed. The author's printed parts only include part of the passive components as part of the sensing and actuation in the corresponding proof of concept examples. However, what has been demonstrated so far is incapable of printing what is typically referred to as an “embedded sensor or actuator”. The demonstrated system in the manuscript relies on external drivers and electronics (which are not embedded). While what is demonstrated is a step forward as an integrated passive mechanical component and indeed is a step closer to

creating a highly complex structure, it is far from achieving “integrated sensing and actuation” as what seems to be implied with what the natural organism has achieved.

We agree with the referee and have adapted the text. Instead of speaking of “integrated sensing and actuation”, we now better highlight the ability to create “integrated sensing and actuation **channels**”.

Line 19ff: It remains an open challenge to fabricate functional systems automatically and quickly with a wide range of elastic properties, resolutions, and integrated actuation and sensing channels [2, 3].

Line 42ff: [...] Hybrid systems that are made of soft, compliant materials [13] but contain rigid, load-bearing parts [14] can resemble natural organisms at the macroscopic scale [15]. Recently developed hybrid soft-rigid systems can already outperform rigid systems in certain unstructured environments [16,17] by adapting to unknown situations [18] and interacting with living beings in a safe manner [19]. In addition, we must include channels and cavities to carry, for example, signals, power, or materials. These features are important but difficult to replicate.

Remark 2E

4. The comparison with the direct ink writing (DIW) approach may not be an apple-to-apple comparison. DIW and Material Jetting are distinct classes of additive manufacturing technologies that have their unique attributes. It is well-known that material jetting can have higher throughput and achieve better multi-material integration but is usually limited with inks with similar chemical basis. Based on the revised manuscript, while a significant advances over the previous material jetting system, it still lacks the versatility of the range of materials that a DIW has shown due to the fundamentally limited compatibility of the ink for the jetting process (e.g., orders of magnitude narrower viscosity range). While contrasting the advancement made in this work with material jetting makes sense, the current comparison with DIW, (which seems to suggest that the proposed system will supplant a DIW system) should be clarified; or further justification and demonstration of how the system can overcome these fundamental limitations in comparison to DIW should be demonstrated.

We thank the referee for the remark on DIW and adapted the sentences to clarify that DIW can operate on a wider set of inks because it is not limited to low viscosity materials like methods that jet ink.

Line 60f DIW methods support a **range of resin viscosities**, [...]

Line 62f Traditional 3D inkjet printing uses thousands of individually addressable nozzles to deposit **low-viscosity** resins [...]

Remark 2F

5.(Minor comment)

Some of the descriptions are vague and can be quantified further. For example, the authors stated that:

"A rapid and accurate multimaterial additive manufacturing method is required to repeatably produce hybrid soft rigid systems with a fine resolution at scale."

It will be much more meaningful if the authors can provide a quantitative description of what they mean by rapid, accurate, hybrid, and fine resolution.

We thank the referee for this remark. We now provide a quantitative description for the words “accurate” and “rapid” in this sentence. Accurate and fine are the same description, so we only provide quantitative info after the word “accurate”. The term hybrid refers to the multi-material combination of soft and rigid materials in a complex system such as the ones presented in this work.

Line 53ff An accurate (few tens of μm) and rapid (millions of voxels/s) multi-material additive manufacturing method is required to repeatably produce hybrid soft rigid systems with a fine resolution at scale.

Remark 2G

6 (Minor comment)

The revised manuscripts have improved their presentations and have clearer descriptions and clearer illustrations. However, a few typos and inconsistencies remain. For example, in the legend of Figure 2, some are labeled as thiol-ene, and some are labeled as thiolene. Some of the character '1' is bolded. While the font size and format meet the minimum requirement of the journal, the graph of Figure 2 can be difficult to read due to the choice of font, the color of the marker, and alignment. While it is readable, it seems to have room for improvement when comparing the quality of the figure with other papers published in this journal.

We are glad the referee sees an improvement in our presentation. We now corrected more typos and inconsistencies. The font and font size were also improved in Figure 2.

Line 479ff:

Fig. 2 | Material characteristics compared to the state-of-the-art. a, Schematic of 3D material jetting. Jetting describes a process where ink in a nozzle is ejected in the form of droplets onto a substrate using a piezoelectric transducer **b**, Acrylate chemistry uses functional monomers cured via radical polymerization (represented by molecules A, B, and C) in their inks. Chain growth polymerization results in a random and irregular distribution of monomers along the polymer backbone (blue). Since all acrylate polymers, rigid and soft, contain this common backbone chain (blue), the properties are controlled by the monomer side groups, R1 and R2, and both the structure and degree of crosslinks R3 (yellow). This combination of structural features typically results in a viscoelastic polymer. **c**, Vision controlled jetting has enabled printing of a step growth polymer (e.g., thiol-ene) using radical polymerization. The polymer chain consists of alternating thiol (D or F) and ene (E or G) monomers. The properties of the polymer are defined by the regular backbone structure and the degree of crosslinking. **d**, The change in elastic modulus during outdoor weathering for Tango Black Plus and soft thiol-ene. Error bars indicate one standard deviation from the mean over four samples. **e**, Change in breakdown strength during outdoor weathering for Tango Black Plus, Agilus 30, and soft thiol-ene. Error bars indicate one standard deviation from the mean over three samples. **f**, The hysteresis of the material is defined by the area enclosed by a stress-strain cycle. The two

acrylates tested showed 3- or 4.43-fold increased hysteresis. g, A dynamic mechanical analysis (DMA) was conducted on soft thiol-ene and two acrylates. Changes in storage modulus were significant between $-35\text{ }^{\circ}\text{C}$ to $-18\text{ }^{\circ}\text{C}$, whereas the two acrylate samples showed significant changes in storage modulus in a much wider range between $-30\text{ }^{\circ}\text{C}$ to $15\text{ }^{\circ}\text{C}$.

Sincerely,

Authors